



# Radical chemistry at a UK coastal receptor site – Part 1: observations of OH, HO₂, RO₂, and OH reactivity and comparison to MCM model predictions

Robert Woodward-Massey[1], Roberto Sommariva[2,4], Lisa K. Whalley[1,3], Danny R. Cryer[1], Trevor Ingham[1], William J. Bloss[4], Sam Cox[5,a], James D. Lee[6,7], Chris P. Reed[6,b], Leigh R. Crilley[4,c], Louisa J. Kramer[4,d], Brian J. Bandy[8], Grant L. Forster[9], Claire E. Reeves[8], Paul S. Monks[2], and Dwayne E. Heard[1]

[1]School of Chemistry, University of Leeds, Leeds, LS2 9JT, UK
[2]School of Chemistry, University of Leicester, University Road, Leicester, LE1 7RH, UK
[3]National Centre for Atmospheric Science, University of Leeds, Leeds, LS2 9JT, UK
[4]School of Geography, Earth and Environmental Sciences, University of Birmingham, Birmingham, B15 2TT, UK
[5]Research Software Engineering Team, University of Leicester, Leicester, LE1 7RH, UK
[6]Wolfson Atmospheric Chemistry Laboratories, Department of Chemistry, University of York, York, YO10 5DD, UK
[7]National Centre for Atmospheric Science, University of York, York, YO10 5DD, UK
[8]Centre for Ocean and Atmospheric Sciences, School of Environmental Sciences, University of East Anglia, Norwich, UK
[9]National Centre for Atmospheric Science, University of East Anglia, Norwich, NR4 7TJ, UK
[a]now at: Digital Research Service, University of Nottingham, Nottingham, NG7 2RD, UK
[b]now at: Faculty for Airborne Atmospheric Measurements, Cranfield University, Cranfield, MK43 0AL, UK
[c]now at: Department of Chemistry, York University, Toronto, M3J 1P3, Canada
[d]now at: Ricardo Energy & Environment, Harwell, Oxfordshire, OX11 0QR, UK

*Correspondence to*: Lisa K. Whalley (l.k.whalley@leeds.ac.uk) and Dwayne E. Heard (d.e.heard@leeds.ac.uk)

**Abstract.** OH, HO₂, total and partially-speciated RO₂, and OH reactivity ($k'_{OH}$) were measured during the July 2015 ICOZA (Integrated Chemistry of OZone in the Atmosphere) project that took place at a coastal site in North Norfolk, UK. Maximum measured daily OH, HO₂, and total RO₂ radical concentrations were in the range $2.6$–$17 \times 10^6$, $0.75$–$4.2 \times 10^8$, and $2.3$–$8.0 \times 10^8$ molecule cm⁻³, respectively. $k'_{OH}$ ranged from 1.7 to 17.6 s⁻¹ with a median value of 4.7 s⁻¹. ICOZA data were split by wind direction to assess differences in the radical chemistry between air that had passed over the North Sea (NW–SE sectors) or major urban conurbations such as London (SW sector). A photostationary steady-state (PSS) calculation underpredicted daytime OH in NW–SE air by ~35%, whereas agreement (~15%) was found within instrumental uncertainty (~26% at 2σ) in SW air. A box model using MCMv3.3.1 chemistry was in better agreement with the OH measurements, but it overpredicted HO₂ observations in NW–SE air in the afternoon by a factor of ~2–3, although slightly better agreement was found for HO₂ in SW air (factor of ~1.4–2.0 underprediction). The box model severely underpredicted total RO₂ observations in both NW–SE and SW air by factors of ~8–9 on average. Measured radical and $k'_{OH}$ levels and measurement-to-model ratios displayed strong dependences on NO mixing ratios. The PSS calculation could capture OH observations at high NO but underpredicted the observations at low NO. The box model overpredicted HO₂ concentrations at low NO in NW–SE air, whereas in SW air, the measurements and model results were in agreement across the full NO range. The box model underpredicted total RO₂ at all



NO levels, where the measurement-to-model ratio scaled with NO. This trend has been found in all previous field campaigns in which total $RO_2$ was measured using the $RO_xLIF$ technique and suggests that peroxy radical chemistry is not well understood under high $NO_x$ conditions.

## 1 Introduction

The removal of trace gases in the troposphere is dominated by reactions with the hydroxyl radical (OH) during the daytime. At semi-polluted locations, OH formation is mainly initiated by the photolysis of ozone ($O_3$) and nitrous acid (HONO):

$$O_3 + hv \ (\lambda < 340 \text{ nm}) \rightarrow O(^1D) + O_2 \tag{R1a}$$
$$O(^1D) + H_2O \rightarrow OH + OH \tag{R1b}$$
$$O(^1D) + M \rightarrow O(^3P) + M \tag{R1c}$$
$$HONO + hv \ (\lambda < 400 \text{ nm}) \rightarrow OH + NO \tag{R2}$$

The OH oxidation of volatile organic compounds (VOCs) in the presence of oxygen results in the formation of organic peroxy radicals ($RO_2$):

$$OH + RH + O_2 \rightarrow RO_2 + H_2O \tag{R3}$$

$RO_2$ radicals may also be formed from the photolysis of oxygenated VOCs (OVOCs). In the presence of NO, $RO_2$ radicals produce hydroperoxyl radicals ($HO_2$) and carbonyl species:


$$RO_2 + NO \rightarrow RO + NO_2 \tag{R4a}$$
$$RO + O_2 \rightarrow R'CHO + HO_2 \tag{R4b}$$

$HO_2$ also reacts with NO to reform OH:


$$HO_2 + NO \rightarrow OH + NO_2 \ , \tag{R5}$$

thus completing the atmospheric reaction cycle known as the hydrogen oxide ($HO_x = OH + HO_2$) cycle. $HO_2$ is also formed by formaldehyde (HCHO) photolysis and by reaction of OH with CO and HCHO. Non-photolytic sources of radicals include
nitrate radical ($NO_3$) chemistry and the ozonolysis of alkenes. Chlorine atoms may also react with VOCs to generate $RO_2$ radicals. The subsequent photolysis of $NO_2$ formed in reactions (R4a) and (R5) results in the production of ozone in the troposphere:



$$\text{NO}_2 + h\nu\ (\lambda < 400\ \text{nm}) \rightarrow \text{O}(^3\text{P}) + \text{NO} \tag{R6a}$$

$$\text{O}(^3\text{P}) + \text{O}_2 + \text{M} \rightarrow \text{O}_3 + \text{M} \tag{R6b}$$

The short lifetimes of OH, $HO_2$, and $RO_2$, collectively known as $RO_x$, make them ideal species to test our understanding of tropospheric oxidation chemistry, particularly when measurements of OH reactivity (the inverse of the OH lifetime, $k'_{OH}$) are also available.

The marine boundary layer (MBL) accounts for a substantial fraction (71%) of the planetary boundary layer. Field measurements of OH and $HO_2$ radicals in the MBL have shown that in general, models are capable of simulating the observed concentrations to within ~30% or better. The majority of these studies were characterised by clean air masses with very low to relatively low NO mixing ratios (Sommariva et al., 2004; Heard et al., 2006; Mao et al., 2009; Whalley et al., 2010; Beygi et al., 2011; Vaughan et al., 2012; Mallik et al., 2018), where observed OH and $HO_2$ concentrations were generally in the range

~3–8 × $10^6$ molecule cm$^{-3}$ and ~1–4 × $10^8$ molecule cm$^{-3}$, respectively. In the MBL, $HO_x$ radical production is normally dominated by the reaction of O($^1$D) with water vapour (reaction (R1)), but HCHO photolysis to $HO_2$ is often an important primary radical source (Ren et al., 2008; Stone et al., 2018). Similarly, owing to low primary VOC levels, OVOCs other than HCHO can account for a significant proportion of OH reactivity (Sommariva et al., 2006; Mao et al., 2009; Whalley et al., 2010; Stone et al., 2018) , and can also be important radical sources. $HO_x$ chemistry was shown to be sensitive to halogen

chemistry in some studies (Bloss et al., 2005b; Whalley et al., 2010; Stone et al., 2018), particularly with regard to the partitioning between OH and $HO_2$ since BrO and IO radicals act to convert $HO_2$ to OH (Sommariva et al., 2006). Heterogeneous uptake of $HO_2$ on aerosols can be a significant $HO_x$ loss route under low NO conditions (Sommariva et al., 2004; Sommariva et al., 2006; Whalley et al., 2010; Stone et al., 2018), but considerable uncertainty surrounds the treatment of heterogeneous processes such as the parameterisation of uptake coefficients ($\gamma_{HO_2}$) (Song et al., 2020), which historically

have sometimes been set to unrealistically high values to achieve measurement-model agreement.

To the authors' knowledge, there are no reported field campaigns in the MBL in which OH, $HO_2$, $RO_2$, and $k'_{OH}$ were all measured simultaneously. Similarly, there are only two studies (Novelli et al., 2014a; Mallik et al., 2018) in the MBL in which OH measurements made by laser-induced fluorescence (LIF) used a technique that allows for the discrimination of OH measurement interferences (Mao et al., 2012). The interferences may arise from both known (e.g., stabilised Criegee

intermediates (Novelli et al., 2014b), $NO_3$ radicals (Fuchs et al., 2016)) and unknown species. In this work, we present interference-free measurements of OH (Woodward-Massey et al., 2020) alongside $HO_2$, total and partially-speciated $RO_2$, and $k'_{OH}$ from a field campaign at a UK coastal receptor site. This complete suite of $RO_x$ measurements allowed for more comprehensive testing of our understanding of MBL chemistry through comparisons to the predictions of a box model. The field campaign took place at a site subject to a variety of air mass types, at which previous field campaigns with (incomplete)

$RO_x$ and/or $k'_{OH}$ measurements were conducted in 1994–1995 (Forberich et al., 1999; Grenfell et al., 1999; Penkett et al., 1999), 2002 (Fleming et al., 2006; Green et al., 2006), and 2004 (Smith, 2007; Lee et al., 2009b).





## 2 The Integrated Chemistry of OZone in the Atmosphere (ICOZA) project

The ICOZA field campaign focussed on the chemistry surrounding the production of ozone, which is harmful to human health (Jerrett et al., 2009), damages vegetation (Krupa et al., 1998), and is a potent greenhouse gas (Ipcc, 2014). The ICOZA
campaign took place in June–July 2015 at the Weybourne Atmospheric Observatory (WAO), which is a Global Atmospheric Watch (GAW) regional station run by the University of East Anglia (UEA) on behalf of the National Centre for Atmospheric Science (https://weybourne.uea.ac.uk/). As shown in Figure 1, the site is located on the North Norfolk Coast, UK (52°57'02" N, 1°07'19"E), ~50 km NNW of Norwich and ~190 km NE of London. The site is situated 16 m above sea level and is surrounded by grass fields on three sides, with the fourth facing due north towards a gently-sloped pebble beach. The nearest
major road is a rural road (A147) located ~800 m to the south.

As the site is situated on the North Sea coast, it is subject to clean air masses that have travelled over the North Sea and originating from polar regions, as well as more polluted air that has been influenced by emissions from major UK cities (e.g., London, Birmingham) ~12–24 h before arriving at the site (Lee et al., 2009b). Polluted continental air, containing aged (by up to 36 hours) anthropogenic emissions from continental Europe, may also be sampled (Lee et al., 2009b). In addition, the site
is subject to emissions from local roads, as well as shipping influences (Cárdenas et al., 1998).

The campaign began on 1st July 2015, but radical measurements commenced on the afternoon of 29th June. The last radical observations were made during the early morning of 22nd July, before the campaign end date of 31st July.

### 2.1 Instrumentation

A list of the instrumentation involved in measurements of trace gases, aerosols, and photolysis frequencies during the ICOZA
campaign is given in Table 1. Instruments sampled ambient air from a height of ~4 m from the roofs of individual shipping containers (Universities of Leeds and Leicester), a van (Birmingham), and from either the roof (~5 m) of the main WAO building directly, or via a common glass manifold (glass, ~15 cm ID) located on a tower that reached ~10 m above the roof. Comparisons of $NO_x$ observations, measured using multiple instruments, indicated no significant heterogeneity in the air sampled from different positions of the site.

### 2.1.1 The Leeds ground-based FAGE instrument

OH, $HO_2$, and $RO_2$ radicals were measured using the fluorescence assay by gas expansion (FAGE) technique (Hard et al., 1984; Heard and Pilling, 2003). The University of Leeds ground-based FAGE instrument, which has been described in detail elsewhere (Creasey et al., 1997; Whalley et al., 2010; Whalley et al., 2013; Whalley et al., 2018; Woodward-Massey, 2018; Woodward-Massey et al., 2020), has been deployed in ~25 intensive field campaigns since 1996 in a variety of chemical
environments. The instrument was used to measure OH and $HO_2$ during a previous campaign at the WAO, namely the Tropospheric ORganic photoCHemistry experiment (TORCH) 2 campaign (Smith, 2007).


Only a brief description of the instrument is given here. Ambient OH concentrations are measured using laser-induced fluorescence (LIF) spectroscopy. The inlet consists of a conical turret (4 cm length, 3.4 cm ID) with a 1.0 mm diameter pinhole through which ambient air is sampled at ~7 slm. The turret is mounted on top of a stainless steel fluorescence cell ($HO_x$ cell),

which is held at ~1.5 Torr (~2 hPa) using a combination of a Roots blower (Leybold RUVAC WAU 10001) and a rotary pump (Leybold SOGEVAC SV2000). A wavelength tuneable solid-state laser (YAG pumped Ti Sapphire laser) with a pulse repetition frequency of 5 kHz is tuned to the OH $A^2\Sigma^+$ (v' = 0) ← $X^2\Pi_{3/2}$(v'' = 0) electronic transition at $\lambda = 308$ nm. Approximately 10–20 mW of laser light is supplied to the fluorescence cell using an optical fibre. OH fluorescence near 308 nm is detected with a micro-channel plate photomultiplier (MCP, Photek PMT325/Q/BI/G with 10 mm diameter photocathode),

which is used together with a 50 ns gating unit (Photek GM10-50) and a 2 GHz 20 dB gain amplifier (Photek PA200-10). Due to failures of the MCP detectors used during ICOZA, channel photomultiplier (CPM, Perkin Elmer 993P) detectors were sometimes used for the $RO_x$ fluorescence cell (see below). Fluorescence signals from the MCP/CPM detectors are analysed using gated photon counting.

HO$_2$ is detected after conversion to OH by the addition of NO (BOC, 99.95%) delivered using a mass flow controller

(MFC, MKS Instruments 1179A series). An advantage for ICOZA relative to TORCH 2 and other previous field campaigns is the addition of instrumental capability for observations of $RO_2$ radicals, using the $RO_x$LIF technique (Fuchs et al., 2008), as well as interference-free measurements of $HO_2$ (Fuchs et al., 2011; Whalley et al., 2013). The Leeds group first applied the $RO_x$LIF method to ambient $RO_2$ observations in London (Whalley et al., 2018) and has since deployed this approach in Beijing (Slater et al., 2020; Whalley et al., 2020). The $RO_x$LIF method relies on the reactions of $RO_2$ radicals with NO (BOC, 500

ppmv in $N_2$) and CO (BOC, 5% in $N_2$) in a flow tube held at ~30 Torr, which result in initial conversion of $RO_2$ to OH ($RO_2$ + NO → $HO_2$, $HO_2$ + NO → OH) and then to $HO_2$ (OH + CO → $HO_2$; very rapid conversion back to $HO_2$ results in minimal radical wall losses) that is finally detected as OH via addition of NO inside a second FAGE cell ($RO_x$ cell) that the $RO_2$ flow tube is coupled to. During fieldwork, the two FAGE cells are used to make sequential measurements of OH, $HO_2$, $HO_2^*$ ($HO_2$ plus an interference from $RO_2$ radicals derived from long-chain alkanes and alkene and aromatic species; see Whalley et al.

(2013) for full details), and total $RO_2$ in the following data acquisition cycle: (1) the first cell ($HO_x$) measures OH while simultaneously the second cell ($RO_x$) measures $HO_2^*$ (high NO flow, 50 sccm; $RO_2$ interference maximised), (2) the $HO_x$ cell measures $HO_2$ (low NO flow, 5 sccm; $RO_2$ interference minimised) while the $RO_x$ cell measures total $RO_2$. The $RO_x$LIF method allows for the speciation of total $RO_2$ into "complex" ($cRO_2$) and "simple" ($sRO_2$) $RO_2$ types (Whalley et al., 2013; Tan et al., 2017; Whalley et al., 2018). $cRO_2$ are those that readily convert to OH in $HO_2^*$ mode ($cRO_2 = HO_2^* - HO_2$; note

that in other previous studies, $cRO_2$ have also been labelled as $RO_2^\#$ or $RO_2$i), and correspond to $RO_2$ radicals derived from alkenes, aromatics, and long-chain (> $C_3$) alkanes. $sRO_2$ concentrations are derived from the difference between total $RO_2$ and $cRO_2$ and correspond to $RO_2$ radicals derived from small-chain (< $C_4$) alkanes. For more details of the speciation of $sRO_2$ and $cRO_2$, the reader is referred to Whalley et al. (2013) and Whalley et al. (2018).

Background signals are normally obtained by scanning the laser wavelength to a position that is off-resonance from the

OH transition line. In the case of OH, this yields the measurement referred to as OHwave (Mao et al., 2012). Alternatively, the



OH background may be determined chemically, via addition of an OH scavenger (e.g., propane) prior to FAGE sampling, which results in an OH measurement known as OHchem (Mao et al., 2012). The recording of OHchem can be used to test for the presence of interferences in conventional OHwave detection. Prior to the ICOZA campaign, an inlet pre-injector (IPI) module (Novelli et al., 2014a) was constructed to facilitate OHchem measurements in the Leeds FAGE system (Woodward-

Massey et al., 2020). The IPI module was first deployed for ambient measurements of OHchem during ICOZA. To test for interferences, two IPI sampling periods were conducted in the middle of the campaign, separated by a few days (3rd–8th July and 12th–16th July). A comparison of OHwave (corrected for the small and well-characterised interference from $O(^1D) + H_2O$, with $O(^1D)$ deriving from laser photolysis of $O_3$ (Woodward-Massey et al., 2020)) and OHchem measurements yielded a slope (OHwave vs OHchem) of 1.16 ± 0.06, with the non-unity value suggesting the presence of a small unknown OHwave

interference during ICOZA on the order of 10–20%, which is smaller than the overall measurement accuracy of 26% at $2\sigma$. The OH data presented in this work correspond to OHchem when such data were available, but OHwave otherwise, where all OHwave data have been corrected for the known interference from $O_3/H_2O$. No attempt has been made to correct the OHwave data for the presence of other unknown interferences, which must be considered as an additional uncertainty in our analyses.

The Leeds FAGE instrument was calibrated by supplying known radical concentrations to the instrument inlets. Radicals

were delivered in an excess flow (~40 slm) of humidified synthetic air (BOC, BTCA 178) using a turbulent flow tube. OH and $HO_2$ were generated in a 1:1 ratio (Fuchs et al., 2011) by the photolysis of water vapour at 184.9 nm using a Hg(Ar) pen-ray lamp (LOT LSP035). For $RO_2$ calibrations, $CH_4$ (BOC, CP grade 99.5%) was added to form $HO_2$ and $CH_3O_2$ in a 1:1 ratio. To enable the calculation of radical concentrations, $N_2O$ (BOC, medical grade 98%) chemical actinometry (Edwards et al., 2003; Faloona et al., 2004) was performed before and after the campaign in order to determine the product of the lamp flux at

184.9 nm and the photolysis time in the flow tube. Multipoint calibrations were performed for all radical species at regular intervals during the campaign, approximately once per week. The calibration factors (i.e., sensitivities) obtained did vary somewhat due to instrumental issues, namely the need to switch between MCP/CPM detectors. One calibration factor was applied to periods in which an MCP was used, and one for periods in which a CPM was used, where both calibration factors were derived from the average sensitivities of multiple MCP or CPM calibrations. As a consequence of the detector changes,

limits of detection (LODs) also varied over the course of ICOZA, with campaign-median 5 min LODs (± 1σ) of $(6.1 \pm 4.1) \times 10^5$, $(4.0 \pm 2.7) \times 10^6$, and $(5.0 \pm 1.2) \times 10^7$ molecule cm$^{-3}$ for OH, $HO_2$, and total $RO_2$, respectively, for a signal-to-noise ratio (SNR) of 2.

### 2.1.2 The Leeds OH reactivity instrument

The Leeds shipping container also housed an instrument used for the measurement of total OH reactivity, $k'_{OH}$. Full details

may be found in Cryer (2016) and Stone et al. (2016) but the key features are described here. The instrument consists of an atmospheric pressure flow tube (85 cm length, 5 cm ID) coupled to an OH fluorescence cell, which was located on the roof of the Leeds container during the ICOZA field campaign. The low pressure in the fluorescence cell (~2 Torr) is provided by the same pumping system as the FAGE cells. The flow tube samples air (via ½" PFA tubing) from close to the FAGE inlets at ~16



slm using a vacuum pump (Agilent Technologies IDP-3 Dry Scroll Pump). The laser flash photolysis pump and probe

technique is used here (Jeanneret et al., 2001; Sadanaga et al., 2004), which involves the 266 nm laser (Quantel USA CFR 200) photolysis (pump) of ambient $O_3$ to generate OH via the reaction of O($^1$D) with $H_2O$. The OH signal decay is then observed in real time by LIF (the probe). Fitting of the first-order exponential obtained yields $k'_{OH}$, after subtraction of the physical decay rate controlled by non-chemical losses of OH (e.g., diffusion). 308 nm probe light is generated using the laser system described above. Previously, OH reactivity was measured at the WAO during TORCH 2 using a different method, the

sliding injector technique with FAGE detection of OH (Ingham et al., 2009; Lee et al., 2009b).

### 2.1.3 Supporting measurements

Formaldehyde (HCHO) was measured using an LIF instrument developed in Leeds, full details of which may be found in Cryer (2016). The instrument is based on the design of Hottle et al. (2009) and uses a pulsed (300 kHz) tuneable fibre laser (TFL3000, Novawave) to generate UV radiation at 353.370 nm, which excites the HCHO $5_{0,5} \leftarrow 5_{1,4}$ rotational transition of

the 4 $A^1A_2 \leftarrow X^1A_1$ vibronic band. As with FAGE, gas is sampled into a low-pressure detection cell (110–120 Torr), but broadband fluorescence is collected at red-shifted wavelengths ($\lambda \sim 390$–550 nm). The fluorescence detected using a PMT (Sens-Tech P25PC photodetector module) and the signal recorded by gated photon counting (PMS400A, Becker and Hickl). The background is determined by moving the laser wavelength to an offline position ($\lambda = 353.360$ nm). The compact HCHO instrument is housed in a shock-insulated 19" rack inside a plastic case, which was situated in the main WAO building during

ICOZA and sampled air through the common glass manifold at a height of ~15 m. HCHO was also measured using Hantzsch colourimetry (Nash, 1953), with reasonably good agreement between the two techniques as demonstrated by the fit $[HCHO]_{LIF}$ = 1.2 × $[HCHO]_{Hantzch}$ + 0.3 ppbv ($R^2 = 0.77$; data not shown).

Photolysis frequencies ($J$) for a variety of species, including $O_3 \rightarrow$ O($^1$D), $NO_2$, $H_2O_2$, HONO, $HNO_3$, and HCHO, were calculated using the actinic flux measured using a $2\pi$ spectral radiometer and published absorption cross-sections and

photodissociation quantum yields; $J$(O$^1$D) was also measured using a $2\pi$ filter radiometer (Meteorologie Consult GmbH) (Bohn et al., 2008). A variety of other supporting instruments (Table 1) were brought to the WAO site. Observational data were also provided by instruments permanently located at WAO (e.g., CO, $NO_x$, $O_3$, $SO_2$, VOCs, meteorological data; see Table 1).

### 2.2 Model description

In this work, radical concentrations were compared to the predictions of a zero-dimensional box model incorporating a kinetic and photochemical mechanism, the Master Chemical Mechanism (MCM, http://mcm.leeds.ac.uk/MCM)(Saunders et al., 1997; Jenkin et al., 2003; Saunders et al., 2003). The current version of the MCM was used, v3.3.1 (Jenkin et al., 2015). The MCM is a near-explicit chemical mechanism, which represents the oxidative degradation of methane and 142 primary emitted VOCs and incorporates ~17 000 reactions of ~6700 closed shell and free radical species. A subset of the MCM with 4258 species



and 12 851 reactions was used instead of the full MCM, reflecting the suite of VOC measurements during ICOZA (Table 1),
e.g., no measurements of $> C_6$ alkanes, and limited BVOC observations (discussed below).

The MCM model simulations were conducted using AtChem2 (https://github.com/AtChem/AtChem2; (Sommariva et al., 2020)). Three model scenarios were used for the interpretation of radical observations: MCM-base, MCM-carb, and MCM-hox. The base model, MCM-base, was constrained to all measured trace gases listed in Table 1, with the exception of radical

species (including $NO_3$ radicals, due to limited measurement data for $NO_3$), and OH reactivity, $Cl_2$, HCHO, the sum of methyl vinyl ketone and methacrolein (MVK+MACR), xylenes, monoterpenes, and  dimethyl sulphide (DMS). MCM-carb was additionally constrained to measured carbonyl species (HCHO, MVK+MACR), but was otherwise identical to the base model; MVK and MACR (both $C_4H_6O$, measured as a sum using proton transfer reaction–mass spectrometry (PTR–MS)) were assumed to be present in a 1:1 ratio. Similarly, MCM-hox was the same as the base model but was additionally constrained to

FAGE-measured $HO_2$. In all simulations, the ratio of trimethylbenzene (TMB) isomers (i.e., $C_9$ aromatics, also indistinguishable by PTR–MS) was assumed to be 1:1:1. In all simulations, NO and $NO_2$ were constrained as separate species rather than as total $NO_x$.

Temperature, pressure, and RH were also constrained in the MCM models, along with spectral radiometer measurements of photolysis frequencies: $O_3 \rightarrow O(^1D)$, $NO_2$, HONO, $HNO_3$, $NO_3$, HCHO, CHOCHO, $CH_3CHO$, $CH_3COCH_3$, $CH_3NO_3$,

$C_2H_5NO_3$, $1\text{-}C_3H_7NO_3$, $2\text{-}C_3H_7NO_3$, and $ClNO_2$. For species with more than one photolytic decomposition channel, branching ratios were taken from the MCM, with the exception of CHOCHO (glyoxal, three channels) for which values were corrected with those used in the Tropospheric Ultraviolet and Visible (TUV) radiation model (Madronich, 1992). Photolysis frequencies that were not measured were calculated using the MCM parameterisation, scaled by a factor derived from measured and calculated $J(NO_2)$ to account for cloud cover.

All measurement constraints were used at their original time resolution, as described in Sommariva et al. (2020). First-order physical losses of unmeasured, model-generated intermediates (e.g., unmeasured OVOCs, organic nitrates, peroxides, acids, and alcohols) through dry deposition were taken from Zhang et al. (2003), where an environment of deciduous trees and long grass/crops was assumed, representative of the immediate area around the WAO. The boundary layer height was estimated at 800 m and kept constant for the duration of the simulations. As examples, these constraints lead to deposition velocities of

~6.4, ~2.8, and ~2.3 cm s$^{-1}$, corresponding to first-order deposition lifetimes of ~4, ~10, and ~12 h, for $HNO_3$, $H_2O_2$, and HCHO, respectively. The lifetime of these model-generated secondary products was determined by their first-order loss rates of dry deposition, heterogeneous uptake (see below), and photolysis, and bimolecular reactions (e.g., with OH and Cl atoms).

In addition to dry deposition, physical losses to aerosols (i.e., heterogeneous uptake) were considered in all model scenarios, represented by the following first-order loss rate (Ravishankara, 1997):


$$k'_{\text{loss}} = \omega A \gamma / 4 \,, \tag{E1}$$



where $\omega$ is the mean molecular speed of the species being taken up, $A$ is the aerosol surface area measured using an aerodynamic particle sizer (APS, range: <0.5–20 μm), and $\gamma$ is the aerosol uptake coefficient. Heterogeneous uptake was
considered for the following species: $O_3$, OH, $HO_2$, $H_2O_2$, $HO_2NO_2$, NO, $NO_2$, HONO, $HNO_3$, $NO_3$, $N_2O_5$, $SO_2$, $SO_3$, HCl, Cl, and $ClNO_2$. $\gamma_{HO2}$ was set to a constant value of 0.1 in all model scenarios, the same value used in analyses of the Clean air for London (ClearfLo) campaign (Whalley et al., 2018).

The model was run for 48 hours (spin-up time) then reinitialised with the values of all species at the end of this period and rerun for the whole campaign. This allowed radical species and other reactive intermediates to reach steady-state levels but
prevented the build-up of secondary products. The model output data were averaged to 15 min for the comparisons featured in this work.

OH concentrations can also be calculated using a photostationary steady-state (PSS) approach, which uses field measured quantities only, providing a check on (1) the internal consistency of OH, $HO_2$, and $k'_{OH}$ observations and (2) whether the OH budget can be balanced using measured quantities. PSS OH was calculated using the following equation:


$$[OH]_{PSS} = P_{OH} / k'_{OH} , \qquad (E2)$$

where $k'_{OH}$ is obtained directly from measured OH reactivity; $P_{OH}$ terms accounted for the photolysis of $O_3$ to $O(^1D)$ (R1a) and reaction with water vapour (R1b), photolysis of HONO (R2), reactions of $HO_2$ (R5) with NO and $O_3$, and alkene ozonolysis
reactions:

$$P_{OH} = 2J(O^1D)[O_3]\,f + J(HONO)[HONO] + k_{HO2+NO}[HO_2][NO] + k_{HO2+O3}[HO_2][O_3] + \Sigma^i\,k_{O3+ALKi}[O_3][ALK_i]Y^{OH}_{ALKi} , \qquad (E3)$$

where $f$ is the fraction of $O(^1D)$ atoms that react with $H_2O$ to form OH (~10%), $J(HONO)$ is the spectral radiometer-determined
HONO photolysis rate, and the final term on the right-hand side accounts for the total OH formation from the ozonolysis of each measured alkene (ALK) $i$ with yield $Y^{OH}_{ALKi}$. Rate constants and yields were taken from MCMv3.3.1 (Saunders et al., 1997; Jenkin et al., 2003; Saunders et al., 2003; Bloss et al., 2005a; Jenkin et al., 2015).

### 2.3 Meteorological and chemical conditions encountered during ICOZA

The overall conditions encountered during the ICOZA campaign are summarised by the time series of meteorological (wind
speed and direction, temperature, RH, photolysis frequencies) and chemical (mixing ratios of NO, $NO_2$, CO, HCHO, isoprene, MVK+MACR, $O_3$, HONO) parameters shown in Figure 2, which includes all available measurements at 15 min time resolution for the period 29th June – 22nd July 2015. As shown in Figure 3, the predominant wind sectors were W, SW, and S (i.e., ~180°–270°). In terms of air mass back-trajectories (Cryer, 2016), during ICOZA the WAO site was generally under the influence of Atlantic air, which had been transported over the UK, likely encountering anthropogenic emissions from major conurbations
(e.g., Birmingham, London, Leicester; Figure 1). However, there were some exceptions to this on certain days of the campaign.



For example, at the start of the campaign on 1st July, air that had spent a considerable amount of time over northern mainland Europe was sampled, which coincided with a heatwave (temperature of up to 30°C) and an event where high mixing ratios of ozone were encountered. Similarly, 11th and 16th July were characterised by a strong European influence, while on 9th July the site was subject to air masses originating from the North Sea.

During the ICOZA campaign, wind speeds were relatively strong, with a median of 5.5 m s$^{-1}$ and a maximum of 12.7 m s$^{-1}$, and tended to drop slightly in the morning. Temperatures generally increased through the day from ~15°C before sunrise to ~20°C in the late afternoon, with a campaign maximum of 29.8°C during the heatwave on 1st July. RH varied between ~40–90% and was strongly anticorrelated with temperature. Based on Figure 3, and given that the SW sector corresponds to air that may have been transported over large urban areas (Figure 1), all ICOZA data (from 2nd July onwards) were split into two

categories according to wind direction – SW winds (180°–270°), and all other winds (NW–SE, <165° and >285°) – as shown in Table 2 and Figure 4. It can be seen that temperatures were generally higher (and conversely RH lower) in SW air. In addition, increased cloud cover in SW air is evident from the slightly lower average values of $J(O^1D)$.

Overall, moderate levels of pollution were observed during the ICOZA campaign. For example, the campaign median NO mixing ratio, for periods of overlap with FAGE radical observations, was 160 pptv with a maximum of 4650 pptv (15 min).

NO generally peaked in the morning, with median values of ~500–1500 pptv at 08:00–10:00 Universal Time Coordinated (UTC = GMT = BST − 1), ~100–400 pptv in the afternoon, and <100 pptv at night (Figure 4). On average, NO mixing ratios were almost a factor of 2 higher in SW air than in NW–SE air (Table 2). NO$_2$ exhibited median and maximum levels of 2.2 and 10.4 ppbv, respectively, and followed an inverse diel profile to that of NO, peaking at night at ~3–4 ppbv with an afternoon minimum of ~1–1.5 ppbv. Both NO and NO$_2$ exhibited significant short-term variability (Figure 2).

The highest ozone mixing ratios of ~110 ppbv were observed on 1st July (Figure 2), which as mentioned above, coincided with elevated temperatures. It should be noted that this day, although interesting as a case study, was not characteristic of the general chemical conditions (particularly ozone levels) of the ICOZA campaign and was thus omitted from the wind sector analysis discussed in this paper (this includes Fig. 4 and Figs. 6–14); VOC measurements were also not available at this time. On average, ozone exhibits a classically-expected photochemical diel profile, with a minimum of ~25–30 ppbv around 06:00

UTC and a maximum of ~35–45 ppbv in the afternoon. Due to higher NO levels, O$_3$ mixing ratios were lower in SW air (Table 2).

The diel profile of HCHO for SW air is similar to ozone (Figure 4), which is typical for an environment where HCHO production is largely driven by the photochemical oxidation of VOCs (Ayers et al., 1997; Cryer, 2016), with a diel minimum of ~800 pptv in the late morning and evening, and a maximum around 16:00 UTC in the range ~1000–1800 pptv. The diel

profile of HCHO in NW–SE air is less pronounced, with lower mixing ratios indicating less integrated photochemical processing. The highest HCHO levels of 3990 pptv were observed during the late morning of 4th July, although unfortunately radical and other measurements are not available for this time, owing to instrumental issues caused by a power cut on the preceding night.





Levels of HONO reached a maximum of ~570 pptv during the night that followed the daytime ozone event discussed
previously (1$^{st}$–2$^{nd}$ July, Figure 2). In general, HONO mixing ratios tended to peak after sunset and midnight in NW–SE (~100
pptv) and SW air (~150 pptv), respectively. There is no obvious diel profile in CO measurements, and no clear difference
between NW–SE and SW wind, with median levels of ~90–150 ppbv observed throughout the day but a few short-term spikes
of up to ~420 ppbv. The flat diel profile observed for CO indicates that, for the most part, the WAO site was not strongly
impacted by fresh anthropogenic combustion emissions during the ICOZA campaign.

Isoprene levels were low during ICOZA, with a campaign median mixing ratio of 24 pptv and a maximum of 450 pptv.
Diel profiles of isoprene were similar between NW–SE and SW air, and bear slight resemblance to that expected from biogenic
emissions with a maximum of ~50 pptv in the afternoon/early evening. The isoprene oxidation products MVK and MACR,
measured as a sum using PTR–MS, exhibited significantly higher levels of ~80–200 pptv in SW air compared to ~20–60 pptv
in NW–SE air. PM$_{2.5}$ levels exhibited no clear diel profile, with similar loadings between the two wind sector types.

**3 Results**

**3.1 Radical and OH reactivity observations and comparison to model predictions**

Figure 5 shows the full time series of OH, HO$_2$, and total RO$_2$ radical concentrations as well as OH reactivity (15 min means)
observed during ICOZA, covering the period 29$^{th}$ June – 21$^{st}$ July 2015. Also shown are the MCM-base model results for all
radical species and $k'_{OH}$ for periods in which measurements of all key species used to constrain the model were available, and
the PSS calculated OH concentrations. The radical observations follow their expected photochemical diel profiles, with
maximum levels around solar noon (~12:00 UTC on cloud-free days) and low nighttime concentrations, approximately an
order of magnitude smaller than during the daytime for OH and HO$_2$, and frequently scattered around zero. There was less of
a day-night contrast for total RO$_2$, for which nighttime levels were almost always above the RO$_2$ LOD (~5 × 10$^7$ molecule
cm$^{-3}$ (~2 pptv)). Unlike radical concentrations, OH reactivity does not appear to show any diel pattern, with a median value of
4.7 s$^{-1}$ but frequent spikes of up to ~10–15 s$^{-1}$ (range = 1.7–17.6 s$^{-1}$). OH reactivity values were much higher at the start of the
campaign (i.e., 1$^{st}$–2$^{nd}$ July 2015), due to the aforementioned heatwave event that coincided with the transport of pollution
from northern continental Europe (Cryer, 2016).

Daily maximum OH concentrations were in the range 2.6–17 × 10$^6$ molecule cm$^{-3}$ and 1.8–13 × 10$^6$ molecule cm$^{-3}$ for
observations and PSS calculations, respectively, based on the 90$^{th}$ percentile of the daytime concentrations with daytime
defined as $J(O^1D) > 5 × 10^{-7}$ s$^{-1}$. The MCM-base modelled OH (1.1–14 × 10$^6$ molecule cm$^{-3}$) is discussed in more detail
below. Similarly, daily maximum observed HO$_2$ and total RO$_2$ levels were in the range 0.75–4.2 × 10$^8$ molecule cm$^{-3}$ and 2.3–
8.0 × 10$^8$ molecule cm$^{-3}$, respectively, or 1.0–4.9 × 10$^8$ molecule cm$^{-3}$ and 0.53–2.8 × 10$^8$ molecule cm$^{-3}$ for MCM-base
predictions. It is clear from these features that the PSS calculation can broadly capture the range in daily maximum OH levels,
while the MCM-base model can generally reproduce peak HO$_2$ but significantly underpredicts midday total RO$_2$. Observed





nighttime concentrations were on the order of $1–3 \times 10^5$, $2–3 \times 10^7$, and $1–2 \times 10^8$ molecule cm$^{-3}$ for OH, HO$_2$, and total RO$_2$, respectively (see Figure 6).

    On shorter timescales it can be seen (Figure 5) that the level of agreement is more variable. For example, the PSS calculation tracks OH observations very tightly for extended periods, but severe underpredictions are often found around midday, with smaller but still significant underpredictions on some mornings. The MCM-base predicted OH levels generally

follow changes in the measurements, but with a tendency towards overprediction during the daytime (median ~10%, see below). Similar to the PSS model capture of OH measurements, MCM-base modelled HO$_2$ concentrations show excellent agreement with measurements for much of the campaign (median daytime overprediction of ~3%). However, in contrast to the comparison between OH measurements and the PSS calculations, on other days the HO$_2$ observations were either under- or overpredicted, with roughly equal examples of each. For total RO$_2$ radicals, the level of agreement is poor (median daytime

underprediction of ~80%), where, with a few exceptions (e.g., 14$^{th}$ July), the MCM-base model cannot reproduce temporal changes in RO$_2$ concentrations, and generally cannot capture their magnitudes with any reasonable degree of success, consistent with the discrepancy between the predicted and observed ranges in daily maxima. OH reactivity is almost always underpredicted (daytime median ~35%), with a few examples of short periods where the MCM-base model reactivity matches the observations.

Figure 6 shows the median diel profiles of observed and modelled radical concentrations and OH reactivity, split by wind direction as in Figure 4. All radicals display their characteristic photochemical diel profiles, peaking around midday (albeit with strong day-to-day variability), and their qualitative features (i.e., overall shapes) are generally well-captured by the various model schemes. Smaller but still significant (i.e., above the LOD for each species) concentrations were observed at night that are generally larger than the model predictions.

Measured diel profiles of OH concentrations were similar in NW–SE and SW air, reaching $\sim 2–4 \times 10^6$ molecule cm$^{-3}$, but with slight differences in the shape of their diel profiles. Overall, the models capture the observations reasonably well (i.e., generally within a factor of 2 during the daytime, although this is larger than the measurement uncertainty of 26% at $2\sigma$), with the best agreement seen in SW air for the MCM-hox model (median difference ~15% during the daytime) and the PSS calculation (~20%). None of the models are able to capture the secondary maximum in OH levels in NW–SE air around 16:00

UTC. The PSS calculation underpredicts OH concentrations throughout the day in NW–SE air by ~35%, but tracks the measurements very tightly in SW air with a slight tendency towards underprediction. This suggests missing OH sources (see Equation (E2)), and is discussed in further detail in our companion paper (Woodward-Massey et al., 2022). For OH, differences between the MCM-base and MCM-carb models are only minor (median difference NW–SE ~1%, SW ~2%), with a greater difference seen for the MCM-hox run.

Measured HO$_2$ levels and diel profiles were very similar between the two wind sector types, with peak levels of around $\sim 1–1.5 \times 10^8$ molecule cm$^{-3}$ in the afternoon and nighttime concentrations on the order of $\sim 2 \times 10^7$ molecule cm$^{-3}$. However, the MCM models predict very different behaviour in NW–SE and SW air. In NW–SE air, HO$_2$ levels are significantly underpredicted in the evening by a factor of ~2, but overpredicted by about a factor of ~2–3 in the afternoon. For SW air, HO$_2$





is still significantly underpredicted in the evening but agreement throughout the daytime is fairly reasonable (median ~1%
difference between measured $HO_2$ and MCM-carb), with less substantial afternoon disagreement. In either wind sector type,
there are strong differences, both positive and negative, between the MCM-base and MCM-carb models (range: −50% to
+70%).

Total $RO_2$ observations reached similar maximum concentrations of ~$5 \times 10^8$ molecule $cm^{-3}$ in NW–SE and SW air, but
exhibit different diel profiles. In NW–SE air, $RO_2$ levels peaked sharply just after midday, with concentrations of ~$1–3 \times 10^8$
molecule $cm^{-3}$ in the morning and late afternoon. In SW air, the profile is broader, with concentrations of ~$2–4 \times 10^8$ molecule
$cm^{-3}$ sustained from mid-morning to the afternoon and maximum levels observed around 16:00 UTC. In contrast to OH and
$HO_2$, the level of measurement/model agreement for total $RO_2$ is poor at all times of day, as might be expected based on their
time series comparison (Figure 5). For example, in NW–SE air, the measurement/MCM-base model ratios range from ~2–5 in
the afternoon to almost 40 in the early morning, with an average value of 8. Similar ratios are found in SW air, albeit with
more substantial afternoon disagreement, with an average of 9. The models do capture the general shape of the diel profiles,
not evident from the time series data in Figure 5, although, the models predict small secondary maxima in total $RO_2$ at night,
which is not seen in the measurements; such behaviour was also found in London (Whalley et al., 2018). Constraining the
model to the few measured carbonyls (MCM-carb) or $HO_2$ (MCM-hox) does little to improve the measurement-model
agreement.

OH reactivity exhibits similar behaviour in the two wind sector types, with relatively flat diel profiles and levels of 3–6
$s^{-1}$. In Figure 6, the grey dashed lines correspond to the OH reactivity from measured trace gases only, and the black lines are
the MCM-base model results that include contributions from model-generated intermediates (i.e., unconstrained OVOCs). In
NW–SE air, the model reactivity roughly tracks temporal changes in the measured reactivity (e.g., the afternoon decrease), but
the reactivity is underpredicted by ~34% throughout the day (maximum ~49%). The contribution of model intermediates to
model reactivity is ~35% on average, with an afternoon maximum of ~63%. In SW air, the measured OH reactivity profile is
flatter but is also underpredicted throughout the day by ~37% on average (maximum ~46%). Model intermediates were less
important than in NW–SE air, but accounted for a slightly greater proportion of model reactivity in the afternoon and evening
of up to ~30–40%, compared to ~22% on average.

### 3.2 RO₂ speciation

The RO$_x$LIF technique allows for "simple" (s$RO_2$) and "complex" (c$RO_2$) organic peroxy radicals to be measured separately,
as discussed in Section 2.1.1. RO$_x$LIF observations of speciated $RO_2$ radicals are compared to MCM-base model predictions
in Figure 7. On average, both observed and modelled s$RO_2$ account for ~60–100% of total $RO_2$ radicals. The overall levels of
s$RO_2$ (~$1–3 \times 10^8$ molecule $cm^{-3}$) and c$RO_2$ (~$0–1.5 \times 10^8$ molecule $cm^{-3}$) are similar between each wind sector type. In NW–
SE air, s$RO_2$ and c$RO_2$ display slightly different diel profile shapes, the latter being suppressed in the morning hours. s$RO_2$ are
always significantly underpredicted by the model (average measurement/model ratio ~ 9), whereas there is agreement for c$RO_2$
around ~06:00 and ~18:00 UTC but disagreement overall (average ratio ~ 7). In SW air, both diel profiles are broader and the





degree of underprediction in the afternoon is worse, with average values of ~10 and ~7 for sRO$_2$ and cRO$_2$, respectively. Similar to NW–SE air, agreement is also seen for cRO$_2$ in the early morning in SW air.

Figure 8 shows the daytime breakdown of RO$_2$ species predicted by the MCM-base model, split according to wind direction. The model predicts that the dominant species in both wind sector types was methylperoxy (CH$_3$O$_2$), with contributions of ~58% and ~55% (daytime median) in NW–SE and SW air, respectively. In NW–SE air, the next most important species is HYPROPO2 (CH$_2$(OH)CH(CH$_3$)O$_2$, formed from OH addition to propene) with a contribution of ~9%, followed by acetylperoxy (CH$_3$CO$_3$, ~7%), BUTDBO2 (CH$_2$(OH)CH(O$_2$)CH=CH$_2$, formed from OH addition to 1,3-butadiene, ~2%), and HOCH2CH2O2 (CH$_2$(OH)CH$_2$O$_2$, formed from OH addition to ethene, ~2%). Other RO$_2$ radicals

contribute ~22% in total. In SW air, the contributions are fairly similar: HYPROPO2 ~6%, acetylperoxy ~8%, BUTDBO2 ~2%, and HOCH2CH2O2 ~2%. Other RO$_2$ radicals are slightly more important than in NW–SE air, with a total contribution of ~25%. Isoprene-derived peroxy radicals (with the most important being ISOPBO2 and ISOPDO2) contribute only ~2% and ~5% in NW–SE and SW air, respectively.

### 3.3 Observed and modelled OH vs $J(O^1D)$

Since OH is formed photochemically with ozone photolysis as a major source (Reaction (R1); see (Woodward-Massey et al., 2022)), OH levels are known to display a strong dependence on $J(O^1D)$. Strong correlations between OH and $J(O^1D)$ have been found for many field campaigns in a range of different environments (Ehhalt, 1999; Brauers et al., 2001; Creasey et al., 2003; Rohrer and Berresheim, 2006; Smith et al., 2006; Bloss et al., 2007; Emmerson et al., 2007; Smith, 2007; Furneaux, 2009; Lu et al., 2012; Vaughan et al., 2012; Lu et al., 2013; Tan et al., 2017; Tan et al., 2018b). In the present work, observed

and modelled OH concentrations are plotted against $J(O^1D)$ for NW–SE and SW air in Figure 9. Here, the data were split into 12 bins with the same number of points, to spread data evenly across the full range of $J(O^1D)$. Power law fits ($y = y0 + Ax^{pow}$) were applied to the datasets for comparison of fit parameters to previous work. The fit parameters may be interpreted as follows: (1) the intercept (y0) is a measure of OH production which does not require the presence of light, for example alkene ozonolysis (but which can also occur during the day); (2) the scaling factor (A) reflects the dependence of OH on other species

such as NO$_x$ and VOCs; (3) the power term (pow) represents the combined effects of photolytic processes (Rohrer and Berresheim, 2006). The fit parameters obtained are summarised in Table 3. In NW–SE air, measured OH concentrations exhibit an almost linear dependence on $J(O^1D)$, with a power term of 0.86. The fit to modelled OH exhibits more curvature, close to a square root dependence, and an intercept smaller than that for the measurements by a factor of 10. In SW air, the measured dependence is less linear (power term = 0.67) but the intercept is similar to that for NW–SE air. In contrast to NW–SE air, the

measurement and model curves are similar such that either can describe the data reasonably well. Overall, the power terms are amongst the lower values of previous reports, which ranged from 0.61–1.3 (Stone et al. (2012) and references therein). These power terms are most similar to those found for other coastal campaigns such as at Mace Head, Ireland (0.84; (Smith et al., 2006)) and Finokalia, Crete (0.68; (Berresheim et al., 2003)).





### 3.4 Observed and modelled RO₂ vs HO₂

It can be seen from Figure 5 and Figure 6 that $RO_2$ observations are well correlated with $HO_2$. To assess the strength of this correlation, $RO_2$ is plotted against $HO_2$ in Figure 10 for both measurement and model results, with fit parameters summarised in Table 4. Observed $RO_2$ and $HO_2$ are indeed strongly correlated, with a stronger correlation in SW air ($R = 0.81$ vs $R = 0.63$ in NW–SE air). For NW–SE air, the correlation is much stronger for $sRO_2$ vs $HO_2$ ($R = 0.68$) than $cRO_2$ vs $HO_2$ ($R = 0.37$) (data not shown). The fit slopes suggest that in NW–SE air $RO_2$ and $HO_2$ coexisted in approximately a 1:1 ratio, while this

was closer to 2:1 for SW air. The non-negligible intercepts of $\sim1–2 \times 10^8$ molecule cm$^{-3}$ suggest that there are some $RO_2$ sources that do not result in the concomitant production of $HO_2$, consistent with the time series data in Figure 5, which may be more relevant at night and possibly indicates a contribution from $NO_3$ chemistry. For the model results, the $RO_2$:$HO_2$ ratio was closer to 1:2 in both NW–SE and SW air during the daytime, but much higher ($\sim$12:1) during nighttime. The different slopes for day and nighttime data in the model cases are not seen in the observations. The increased slope for the model results

during nighttime indicates slower $RO_2 \rightarrow HO_2$ cycling due to lower NO levels.

### 3.5 Observed and modelled OH, HO₂, RO₂, and *k'*ₒₕ vs NO

Radical levels are known to display a strong dependence on $NO_x$ concentrations since radical propagation is promoted by NO, and radical loss is often dominated by the reactions of radicals with NO and $NO_2$. In recent studies utilising the $RO_xLIF$ technique, it has become apparent that measurement-model ratios for $RO_2$ are particularly sensitive to NO (Tan et al., 2017;

Tan et al., 2018b; Whalley et al., 2018; Slater et al., 2020; Whalley et al., 2020), which has implications for the calculation of ozone production rates, discussed in more detail in the companion to this paper (Woodward-Massey et al., 2022). The dependence of daytime ($J(O^1D) > 5 \times 10^{-7}$ s$^{-1}$) radical concentrations and OH reactivity values on NO mixing ratios is shown in Figure 11, split according to wind direction. For OH only, both measured and modelled concentrations were normalised to the campaign-average $J(O^1D)$ to remove the dependence on OH source strength (= OH_Jnorm, e.g., (Tan et al., 2017)). This

approach is justified by the almost linear dependence of OH on $J(O^1D)$ (Figure 9); similar trends were also found for un-normalised OH albeit with more scatter (data not shown). The corresponding measurement-model ratios for radical species are shown in Figure 12.

In NW–SE air, observed OH_Jnorm levels exhibit a classically-expected dependence on NO, increasing up to $\sim$100 pptv NO before decreasing at higher NO (Figure 11). The MCM-base model reproduces the measured trend reasonably well.

However, the PSS model significantly underpredicts the observations at low NO, yielding measurement-model ratios of $\sim$2–3 below $\sim$200 pptv NO (Figure 12), which is greater than the estimated combined measurement-model uncertainty ($\sim$50%). In SW air, measured OH_Jnorm decreases with NO across the full NO range. The PSS model underpredicts the observations more severely at low NO (<300 pptv), yielding similar measurement-model ratios to those in NW–SE air. The MCM-base model slightly underpredicts the observations at low NO (by up to $\sim$90%) but there is reasonable agreement (within $\sim$40%) at

moderate to high NO (>300 pptv). The different behaviour for measured OH_Jnorm with respect to NO in NW–SE and SW



air might be expected from the strong differences in scaling factors (A, Figure 9 and Table 3) in the analysis in Section 3.3, which reflect the dependence of OH on species such as $NO_x$ and VOCs.

Measured $HO_2$ in NW–SE air exhibits a weak decreasing trend with NO, with levels of ~0.7–1.3 × $10^8$ molecule cm$^{-3}$ below ~1 ppbv NO and ~0.3 × $10^8$ molecule cm$^{-3}$ above this threshold. In contrast, the model dependence on NO is much

stronger such that $HO_2$ levels are overpredicted by up to a factor of ~3 at low NO. In the highest NO bin, measured $HO_2$ is underpredicted by a factor of ~2. In SW air, both measured and modelled $HO_2$ decrease sharply with NO, from ~2 × $10^8$ molecule cm$^{-3}$ at ~100 pptv NO to ~0.1–0.3 × $10^8$ molecule cm$^{-3}$ above 1 ppbv. For this wind direction, the measurements and model results are in agreement across the full NO range.

Overall, measured OH and $HO_2$ are in reasonable agreement with the PSS calculation and the MCM-base model

prediction, respectively, at high NO. There is also good agreement between measured and MCM-base OH at moderate to high NO. However, $RO_2$ radicals are significantly underpredicted by the base model across all NO mixing ratios in both NW–SE and SW air. Observed and modelled $RO_2$ concentrations display a constant decrease with NO in either wind sector type. Comparing the two sets of observations, the dependence is steeper in SW air. In both wind sector types, the model NO dependence of $RO_2$ is steeper than the corresponding measurement NO dependence, such that the measurement-model ratio

increases from ~2–3 for NO < 100 pptv to ~10–30 for NO > 1 ppbv (Figure 12). Such discrepancies likely relate to the model underprediction of OH reactivity, the degree of which also scaled with NO (Figure 11), since this indicates missing $RO_2$ sources from the OH oxidation of missing VOCs. Missing OH reactivity, i.e., the difference between measured and modelled OH reactivity, reached values of ~2–3 s$^{-1}$ for NO > 1 ppbv, or ~30–45% of measured reactivity.

The increasing underprediction of $RO_2$ radicals as NO increases has been seen in all previous field campaigns in which

$RO_2$ (distinct from $HO_2$) was measured using the $RO_x$LIF technique (Fuchs et al., 2008; Whalley et al., 2013). $RO_2$ measurement-model ratios as a function of NO from these campaigns (Tan et al., 2017; Tan et al., 2018b; Whalley et al., 2018; Slater et al., 2020; Whalley et al., 2020) are compared with ICOZA in Figure 13. It can be seen that the measurement-model discrepancy starts to appear at lower NO (i.e., <100 pptv) for ICOZA in comparison to the other campaigns, although the curves for ICOZA and AIRPRO summer display strong overlap in the ~100–600 pptv NO range. There is also some overlap

between the curves for ICOZA and BEST-ONE (Tan et al., 2018b), a winter campaign conducted at a suburban site near Beijing, at low/moderate NO (~100–200 pptv). Overall, the largest measurement-model ratios were found in London (Whalley et al., 2018) and central Beijing (Slater et al., 2020; Whalley et al., 2020), but at higher NO levels (>10 ppbv) than those seen in most other campaigns including ICOZA.

To further explore the $RO_2$ discrepancy found for ICOZA, the contribution of $sRO_2$ to total $RO_2$ is plotted as a function

of NO for measurement and model results in Figure 14. For the measurements, the $sRO_2$ contribution increases with NO in both NW–SE and SW air from ~0.7 to values close to 1. In contrast, the model predicts a constant $sRO_2$ fraction of ~0.7, in accordance with the dominance of $CH_3O_2$ (Figure 8). The reasons for the strong dependence of the measured $sRO_2$ fraction on NO are unclear, but may be due to the NO-mediated propagation of $cRO_2$ to $sRO_2$ as VOCs are increasingly fragmented into





smaller and less complex $RO_2$ species. Alternatively, $cRO_2$ formation may be facilitated by low $NO_x$ levels, e.g., due to
autoxidation chemistry (Crounse et al., 2013; Jokinen et al., 2014; Bianchi et al., 2019).

### 3.6 Missing $k'_{OH}$ vs OVOCs and temperature

Missing OH reactivity has been found in many previous field studies in which OH reactivity was measured and compared to calculated reactivity or model simulations (Kovacs et al., 2003; Ren et al., 2003; Di Carlo et al., 2004; Sinha et al., 2008; Lee et al., 2009b; Lou et al., 2010; Mao et al., 2012; Nolscher et al., 2012; Edwards et al., 2013; Brune et al., 2016; Whalley et al.,
2016; Kumar et al., 2018). Missing OH reactivity is normally attributed to either unmeasured primary VOCs (e.g., BVOCs), or unmeasured VOC oxidation products (i.e., OVOCs). To test which was responsible for the missing reactivity observed for ICOZA, missing OH reactivity (measured – modelled) was binned against various chemical concentrations and temperature. These data are shown in Figure 15. It can be seen that missing reactivity exhibits strong correlations ($R^2 \geq 0.83$) with several measured OVOCs, such as acetaldehyde, acetone, and methanol (all constrained in MCM-base). This finding suggests that the
missing reactivity is due to unmeasured VOC oxidation products that were not well simulated by the base model. The only OVOCs measured and constrained in the base model were acetone, acetaldehyde, and methanol and as such many OVOCs were missing, e.g., the oxidation products of $>C_2$ VOCs. Weaker correlations ($R^2 \leq 0.7$) were found for isoprene (maximum = 418 pptv) and the PTR-MS measured sum of monoterpenes (maximum = 105 pptv), such that unmeasured primary BVOCs are unlikely to be the root of the missing reactivity. BVOC emissions are known to display an exponential dependence on
temperature (Guenther et al., 1993). It is therefore expected that missing reactivity should scale exponentially with temperature if missing biogenic species are responsible (Di Carlo et al., 2004). As shown in Figure 15, this was not the case for ICOZA and the dependence is clearly linear, albeit over a relatively small temperature range of ~12–24°C. This is further evidence that the missing reactivity for ICOZA is due to OVOCs, not a primary biogenic species. It is hypothesised that the correlation with temperature is due to increased VOC oxidation rates at high temperature that results in greater OVOC production. Missing
reactivity is also reasonably well correlated with toluene ($R^2 = 0.84$, data not shown), such that unmeasured aromatic VOCs could also be responsible, as suggested by Lee et al. (2009b).

When missing OH reactivity is calculated using the MCM-carb model, which is additionally constrained to measured HCHO and MVK+MACR, all the correlations in Figure 15 remain ($R^2 \geq 0.83$), with the exception of temperature ($R^2 = 0.61$). Therefore, species other than HCHO and MVK+MACR must be responsible for the missing OH reactivity. In recent years,
OVOC emissions have increased in importance in the UK, with ethanol now the largest contributor to non-methane VOCs in terms of mass emissions (Lewis et al., 2020). More generally, alcohols are now the largest contributors to ozone production (~30%) in terms of their photochemical ozone creation potentials (POCPs). It is therefore critical that in future field campaigns, alcohols such as ethanol and isopropanol are measured to evaluate their impacts on radical budgets and ozone production.





## 4 Discussion

### 4.1 Comparison to previous coastal field campaigns

OH, HO$_2$, and OH reactivity have previously been measured at the WAO using FAGE, most recently in May 2004 as part of the TORCH 2 campaign (Smith, 2007; Lee et al., 2009b). However, NO levels were generally higher during TORCH-2 (mean 0.62 ppbv, range ~0–50 ppbv) than ICOZA (mean 0.38 ppbv, range ~0–7.5 ppbv), likely due to NO$_x$ emissions reductions over the 11 years that separate the two campaigns (https://www.gov.uk/government/statistics/emissions-of-air-
pollutants/emissions-of-air-pollutants-in-the-uk-nitrogen-oxides-nox). On a diel-average basis, OH peaked at 12:00 UTC with similar levels (~4 × 10$^6$ molecule cm$^{-3}$) to ICOZA. OH has also been measured by DOAS at the WAO, with measured peak daytime levels of ~4–7 × 10$^6$ molecule cm$^{-3}$ (Forberich et al., 1999; Grenfell et al., 1999). For other ground and ship-based campaigns in the MBL, measured noontime OH concentrations were mostly in the range ~4–6 × 10$^6$ molecule cm$^{-3}$, and generally the observations have been found to agree with model predictions to within ~30% on average during the daytime in
the MBL (Sommariva et al., 2004; Sommariva et al., 2006; Whalley et al., 2010; Beygi et al., 2011; Van Stratum et al., 2012). During the NASA airborne Atmospheric Tomography study (ATom), OH concentrations in the MBL were on the order of ~1– 4 × 10$^6$ molecule cm$^{-3}$ and a model was able to reproduce them to generally within 40% (Brune et al., 2020).

Measured HO$_2$ levels were also very similar during TORCH 2, peaking in the afternoon (~14:00 UTC) at ~8 × 10$^7$ molecule cm$^{-3}$ (cf. 1 × 10$^8$ molecule cm$^{-3}$ for ICOZA, Figure 6) (Smith, 2007). For other previous measurements of HO$_2$, the
observed concentrations and levels of measurement-model agreement have generally been quite variable. In terms of HO$_2$, ICOZA is most similar to the North Atlantic Marine Boundary Layer EXperiment (NAMBLEX), which took place at Mace Head, Ireland, in summer 2002 (Heard et al., 2006). During NAMBLEX, noontime HO$_2$ concentrations were in the range 0.9– 2.1 × 10$^8$ molecule cm$^{-3}$ and were overpredicted by up to a factor of 2 (Sommariva et al., 2006). These results are almost identical to the findings of ICOZA despite the substantial differences in chemical conditions (e.g., the much lower
anthropogenic influence and the role of halogen species during NAMBLEX). During the Diel Oxidants Mechanisms In relation to Nitrogen Oxides (DOMINO) campaign in SW Spain in Nov–Dec 2008 (Van Stratum et al., 2012), similar HO$_2$ levels of up to ~1.5 × 10$^8$ molecule cm$^{-3}$ were observed (in continental air, analogous to the SW sector in the present work). However, model calculations were only performed for one day, using a mixed layer model, which showed significant morning and afternoon underpredictions. This may be due in part to interferences from RO$_2$ radicals (Fuchs et al., 2011; Whalley et al.,
2013), which were not known about at the time. For other campaigns, HO$_2$ concentrations were generally above ~2 × 10$^8$ molecule cm$^{-3}$ (Sommariva et al., 2004; Whalley et al., 2010; Beygi et al., 2011), higher than the range observed during ICOZA (~0.5–2 × 10$^8$ molecule cm$^{-3}$), and the observations have mostly been overpredicted. During ATom, HO$_2$ concentrations in the MBL were on the order of ~1–5 × 10$^8$ molecule cm$^{-3}$ and were well captured by a model (agreement within 40%) (Brune et al., 2020).

RO$_2$ radicals were not measured during TORCH 2, but the sum of peroxy radicals (HO$_2$ + RO$_2$) have been measured using peroxy radical chemical amplification (PERCA) during other campaigns at the WAO (Penkett et al., 1999; Fleming et al.,





2006; Green et al., 2006). The most recent campaign, INSPECTRO (Influence of clouds on the spectral actinic flux in the lower troposphere), took place in September 2002 (Green et al., 2006). Peak $HO_2 + RO_2$ levels of ~2–5 × $10^8$ molecule $cm^{-3}$ were observed at or a few hours after solar noon, with nighttime levels on the order of ~5–7 × $10^7$ molecule $cm^{-3}$. In

comparison, $HO_2 + RO_2$ concentrations during ICOZA were fairly similar during the daytime with levels of ~4–6 × $10^8$ molecule $cm^{-3}$ at solar noon (Figure 6), but nighttime concentrations were approximately 3–4 times higher (~2 × $10^8$ molecule $cm^{-3}$). It is possible that during ICOZA, nighttime ozonolysis or $NO_3$ radical chemistry was more active. Nighttime $HO_2$ + $RO_2$ levels of up to ~1 × $10^8$ molecule $cm^{-3}$ were observed using PERCA at Mace Head during the Eastern Atlantic Spring Experiment (EASE 97) in April–May 1997 (Salisbury et al., 2001). During DOMINO 2008, which was heavily influenced by

urban-industrial/petrochemical emissions, $HO_2 + RO_2$ concentrations measured by PERCA were much higher, with daytime maxima of ~2–12 × $10^8$ molecule $cm^{-3}$ and frequent nighttime spikes of up to ~20 × $10^8$ molecule $cm^{-3}$ on nights with air arriving from the industrial sector (Andres-Hernandez et al., 2013). Similarly, during TexAQS (Texas Air Quality Study) 2006, ship-based PERCA measurements in industrial regions around Houston, Texas showed that $HO_2 + RO_2$ levels could reach ~30 × $10^8$ molecule $cm^{-3}$ during nighttime (Sommariva et al., 2011) in highly polluted areas, although, in less polluted areas,

nighttime $HO_2 + RO_2$ levels were on the order ~2–10 × $10^8$ molecule $cm^{-3}$.

The mean OH reactivity measured during TORCH 2 (4.9 $s^{-1}$) is virtually identical to that measured during ICOZA (5.0 $s^{-1}$) (Lee et al., 2009b). The maximum OH reactivity was higher for ICOZA (17.6 $s^{-1}$) than TORCH 2 (9.7 $s^{-1}$) due to the pollution episode at the start of ICOZA (Figure 2). The mean missing OH reactivity in terms of the difference between measured OH reactivity and that calculated using trace gases only was 1.9 $s^{-1}$ (39%) during TORCH 2 (cf. 2.4 $s^{-1}$, 48% for

ICOZA). A box model using MCMv3.1 chemistry (Bloss et al., 2005a) was used to simulate OH reactivity, reducing the missing reactivity to 1.4 $s^{-1}$ or 29% (cf. 1.7 $s^{-1}$, 36% for ICOZA). Lee et al. (2009b) speculated that the missing reactivity may be due to a potentially large number of unmeasured, high molecular weight aromatic compounds, but that this could also be due to missing OVOCs, as we have suggested based on the data in Figure 15. OH reactivity was measured using the comparative reactivity method (CRM) at a Mediterranean coastal receptor site in Corsica, France during summer 2013

(Zannoni et al., 2017). Total OH reactivity reached a maximum of ~17 $s^{-1}$ with an average of ~5 $s^{-1}$, where 44% of the reactivity was missing. OH reactivity was much higher during the DOMINO campaign due to the influence of urban-industrial/petrochemical emissions, with an average value of 18 $s^{-1}$ measured using the CRM (Sinha et al., 2012).

To date, only a handful of studies have measured OH reactivity at coastal locations but there have been airborne campaigns conducted in the marine boundary layer. An airborne OH reactivity instrument was deployed during flights over the Pacific

Ocean for the Intercontinental Chemical Transport Experiment-B (INTEX-B) campaign (Mao et al., 2009). In the boundary layer (i.e., < 2 km altitude), measured OH reactivity was ~4 $s^{-1}$ on average, while that calculated from measured reactants was only ~1.5–2 $s^{-1}$ (i.e., ~50–60% missing), similar to ICOZA. During the NASA Atmospheric Tomography (ATom) campaign involving flights over the Atlantic and Pacific Oceans, measured OH reactivity at < 2 km was on the order of ~2 $s^{-1}$, with missing reactivity on the order of ~0.5–1 $s^{-1}$ (~25–50% missing) (Thames et al., 2020). The authors suggested that, based on





correlations of missing OH reactivity with HCHO, DMS, butanal, and sea surface temperature, there were
       unmeasured/unknown VOCs/OVOCs associated with oceanic emissions, in agreement with our findings.

### 4.2 Differences and similarities between the NW–SE and SW wind sectors

       Differences between the NW–SE and SW wind sectors in terms of radical and OH reactivity levels, and model
       performance, were expected based on the strong differences in chemical conditions between the two wind sector types (Table
2 and Figure 4). However, many aspects are fairly similar between the two wind sector types, for example measured OH, $HO_2$,
       $RO_2$, and OH reactivity levels. Perhaps the most striking difference between the two wind sector types is the model
       performance for OH and $HO_2$ (Figure 6). In NW–SE air, measured OH is underpredicted by the PSS calculation by ~35% on
       average, but reasonable agreement is found in SW air (within 20% on average). Similarly, $HO_2$ is overpredicted by both the
       MCM-base and MCM-carb models by a factor of 2–3 during the afternoon in NW–SE air, but reasonable agreement is found
between measured $HO_2$ and the MCM-carb model for daytime SW air. In contrast, the model underprediction of $RO_2$ is more
       severe in SW air compared to the NW–SE sector, suggesting that the good agreement found for $HO_2$ may be fortuitous (i.e., if
       the model was able to reproduce $RO_2$, then $RO_2 + NO \rightarrow HO_2$ reactions would likely lead to the model overpredicting $HO_2$).
       The underprediction of OH and overprediction of $HO_2$ in NW–SE air, only occurs at low-$NO_x$ (Figure 11 and Figure 12).
       Possible reasons for these discrepancies are discussed in Section 4.4.

### 4.3 Differences between models

       The carbonyls HCHO and MVK+MACR were not constrained in the MCM-base model because of several gaps in the time
       series of these measurements. The MCM-base model performance in simulating these carbonyls was assessed, where it was
       found that there was reasonable agreement for MVK+MACR on a diel average basis, but that HCHO concentrations were
       significantly overpredicted in the afternoon (data not shown). The differences in the calculated concentrations of these OVOC
compounds is the cause of the differences between the MCM-base and MCM-carb simulations of radical species (Figure 6).
       Similarly, $HO_2$ concentrations were generally overpredicted by both the MCM-base and MCM-carb models, and therefore
       constraining to $HO_2$ (MCM-hox model) has impacts on the model OH and $RO_2$ concentrations (Figure 6). Since the MCM-
       base model also underpredicted OH reactivity, there are additional differences between the PSS calculation of OH and the
       MCM model predictions.

For OH, differences between the MCM-base and MCM-carb runs are relatively minor and both positive and negative
       (Figure 6). There are two competing effects here: HCHO is a minor OH sink (~6% (Woodward-Massey et al., 2022)) and thus
       an overprediction of HCHO would lead to lower OH in MCM-base relative to MCM-carb. In contrast, the base model
       overprediction of HCHO leads to a greater $HO_2$ source strength that would drive higher OH levels in MCM-base relative to
       MCM-carb. Similarly, $HO_2$ was overpredicted in MCM-base and MCM-carb such that constraining the model to $HO_2$ (MCM-
hox) resulted in lower model OH levels. Finally, due to the MCM model underprediction of OH reactivity, PSS calculated OH
       levels were lower than the MCM model concentrations. At low NO (Figure 11), the better agreement found for the OH/MCM-





base case compared to the OH/PSS case is driven by the MCM overprediction of $HO_2$, i.e., the agreement for OH does not necessarily mean the OH chemistry is well understood at low NO. Peak afternoon $RO_2$ concentrations were similar for the MCM-base and MCM-carb simulations (Figure 6). However, the reduced OH in MCM-hox results in reduced afternoon $RO_2$
levels.

### 4.4 Overprediction of $HO_2$ under low-$NO_x$ conditions in NW–SE air

In several previous field campaigns, $HO_2$ concentrations were overpredicted under low-$NO_x$ conditions (Sommariva et al., 2004; Sommariva et al., 2006; Kanaya et al., 2007; Griffith et al., 2013; Whalley et al., 2018). Extremely low NO levels of < 3 pptv were observed during the Southern Ocean Photochemistry Experiment (SOAPEX-2), which took place at Cape Grim
in austral summer 1999. $HO_2$ observations were overpredicted by ~40%, but improved agreement could be found by inclusion of $HO_2$ uptake with an uptake coefficient ($\gamma_{HO2}$) of unity (Sommariva et al., 2004). $HO_2$ uptake was considered in the present work, using $\gamma_{HO2} = 0.1$. In our companion paper, we show that even increasing $\gamma_{HO2}$ to unity has little impact on the experimental budget for $HO_2$, thus heterogeneous uptake alone cannot explain the $HO_2$ measurement-model discrepancy under low-$NO_x$ conditions in NW–SE air during ICOZA.

As discussed above, $HO_2$ measurements were overpredicted by a factor of 2 during NAMBLEX, for which the model analysis was performed for days with low $NO_x$ levels (NO < 30 pptv) (Sommariva et al., 2006). Agreement was improved when the model was constrained to measured OVOCs (acetaldehyde, methanol, and acetone in the case of NAMBLEX), similar to the improvement in measurement-model agreement seen for the MCM-carb run (constrained to HCHO and MVK+MACR) in the present work (Figure 6). Additionally, at Mace Head, seaweed beds are exposed at low tide that represent
a significant source of reactive halogen species such as $I_2$ and $CH_2I_2$ (Carpenter et al., 1999; Carpenter et al., 2003; Mcfiggans et al., 2004). Halogen oxides (XO, where X = Br, I) are able to convert $HO_2$ to OH:

$$HO_2 + XO \rightarrow HOX + O_2 \tag{R7}$$

$$HOX + h\nu \rightarrow OH + X, \tag{R8}$$

where hypohalous acids (HOX) may also undergo heterogeneous loss to aerosols. In a steady-state analysis, (Bloss et al.,
2005b) found that up to 40% of $HO_2$ could be lost to IO under low-$NO_x$ conditions, for measured IO levels of 0.8–4.0 pptv (Commane et al., 2011). In the full modelling study (Sommariva et al., 2006), constraining the model to BrO and IO resulted in similar decreases in model $HO_2$, depending on the uptake coefficients used for HOI and HOBr. Reactive iodine species were not measured during ICOZA, and their influence is expected to be negligible due to the lack of seaweed beds at the WAO site. However, it is possible that there was a source of reactive bromine through sea salt aerosol chemistry (Keene et al., 2009). We
therefore speculate that inclusion of reactive halogens could simultaneously reduce the underprediction of OH and the overprediction of $HO_2$ under low-$NO_x$ conditions in NW–SE air. To our knowledge there have been no measurements of $I_2$, BrO, or IO at the WAO (John Plane, personal communication). In our companion paper (Woodward-Massey et al., 2022), we show that incorporation of an artificial species "Y" that converts $HO_2$ to OH (Lu et al., 2012; Lu et al., 2013) does result in slight improvements to the agreement between experimental OH and $HO_2$ production and destruction rates. We also speculate



that if another artificial species "Z" is able to convert $HO_2$ to $RO_2$, this would reduce both the overprediction of $HO_2$ and the underprediction of $RO_2$ at low-$NO_x$ conditions (Figure 11 and Figure 12 in the present work).

OH and $HO_2$ were measured at Rishiri Island, Japan, in September 2003 (Kanaya et al., 2007). Daytime $HO_2$ levels were overpredicted by almost a factor of 2. In addition to halogen chemistry and $HO_2$ uptake, the authors also considered the possibility that $HO_2 + RO_2$ reactions were faster than previously thought. In our companion paper, we show that $RO_2$

destruction rates exceed $RO_2$ production rates at high NO, but the two are equal at low NO. Increasing the rate of $HO_2 + RO_2$ reactions would result in increased $RO_2$ destruction rates, therefore worsening the agreement between $RO_2$ destruction and production rates, which should be in balance. For this reason, we do not think that faster-than-expected $HO_2 + RO_2$ reactions are the cause of the overprediction of $HO_2$ levels under low-$NO_x$ conditions in NW–SE air during ICOZA.

During the Clean air for London (ClearfLo) campaign in summer 2012, $HO_2$ concentrations were overpredicted by a box

model using MCMv3.2 by up to a factor of 10 at low NO (< 1 ppbv) (Whalley et al., 2018). The model $HO_2$ was somewhat reduced but the observations could still not be reconciled after inclusion of both $HO_2$ aerosol uptake (using $\gamma_{HO2} = 1$) and autoxidation chemistry (Bianchi et al., 2019), which is now known to play a significant role in the gas phase oxidation of both BVOCs (Crounse et al., 2011; Ehn et al., 2014; Jokinen et al., 2014; Berndt et al., 2016; Zha et al., 2017) and anthropogenic VOCs (AVOCs) (Wang et al., 2017; Molteni et al., 2018; Mehra et al., 2020; Wang et al., 2020). Whalley et al. (2018) found

that good agreement between the model and $HO_2$ measurements could be found if the rate of $RO_2 + NO \rightarrow HO_2$ propagation was reduced, in their case by reducing the branching ratio for alkyl nitrate formation. In our companion paper, we show that reducing the rate of $RO_2 + NO \rightarrow HO_2$ propagation does help to resolve budget imbalances for $HO_2$ and $RO_2$, even at the lower NO levels observed in the afternoon due to the dominance of $RO_2 + NO$ as an $RO_2$ sink and $HO_2$ source.

### 4.5 Underprediction of RO$_2$ under high-NO$_x$ conditions

Under moderate and high-$NO_x$ conditions (above ~200–300 pptv NO), reasonable measurement-model agreement is found for OH and $HO_2$, i.e., generally to within a factor of 2 (Figure 11 and Figure 12). However, total $RO_2$ concentrations are much more significantly underpredicted, by as much as a factor of ~30 at the highest NO levels encountered (above ~2 ppbv), coincident with increased missing OH reactivity. As shown in Figure 13, this phenomenon has been observed in several other field campaigns (Tan et al., 2017; Tan et al., 2018b; Whalley et al., 2018; Slater et al., 2020; Whalley et al., 2020). Tan et al.

(2017) found that an additional primary $RO_2$ source from chlorine chemistry could explain a small portion (10–20%) of the missing $RO_2$ in their study. Whalley et al. (2018) found that chlorine chemistry increased modelled $RO_2$ for the ClearfLo campaign by ~20% in the morning when $NO_x$ levels were high, in comparison to $RO_2$ underpredictions of greater than factor of 10. Since the major Cl atom precursor $ClNO_2$ was measured during ICOZA (Sommariva et al., 2018) and constrained in all model scenarios, $ClNO_2$ photolysis to form Cl atoms and the subsequent reactions of Cl with VOCs is not thought to be the

source of the missing $RO_2$ in the present study. However, as the chlorine chemistry in MCMv3.3.1 is limited to reactions with alkanes, additional chlorine chemistry (e.g., reactions with alkenes, OVOCs, etc.) may be needed to fully assess the role of chlorine during ICOZA.



Since missing OH reactivity was also found at high $NO_x$ conditions, some of the missing $RO_2$ may be due to the reactions of OH with unmeasured VOCs. However, as we show in our companion paper (Woodward-Massey et al., 2022), evidence is also found for missing $RO_2$ sources at high $NO_x$ using calculations constrained to measured OH reactivity. Thus the missing OH reactivity cannot fully explain the missing $RO_2$. It is possible that the missing $RO_2$ found for ICOZA is not due to a missing $RO_2$ source, but an overestimated $RO_2$ sink. As discussed above, reducing the rate of $RO_2 + NO \rightarrow HO_2$ propagation (i.e., a reduced $RO_2$ sink) helps to resolve budget imbalances for $HO_2$ and $RO_2$, similar to that found for $HO_2$ by Whalley et al. (2018). We found the largest improvement to the experimental budget balance in the morning, when $NO_x$ levels were at their highest. However, the $RO_2 + NO$ rate constant (originally the generic value used in the MCM) had to be reduced by a factor of 5, when laboratory measurements of such rate constants have not reported values as low and have uncertainties in the range ~15–35% (Orlando and Tyndall, 2012).

Recently, Whalley et al. (2020) presented measurements of OH, $HO_2$, $RO_2$, and OH reactivity in summertime Beijing. $RO_2$ concentrations were underpredicted by a box model with MCMv3.3.1 chemistry, most severely at high $NO_x$ (Figure 13). Missing OH reactivity was also identified. The measurement-model agreement for $RO_2$ was significantly improved after the model inclusion of an α-pinene derived $RO_2$ radical, C96O2 (MCM nomenclature), formed at a rate set equal to the level of missing OH reactivity. This complex $RO_2$ species does not generate $HO_2$ directly from its reaction with NO, but instead the RO radical formed preferentially isomerises (via a H-shift) to form another $RO_2$ radical in the presence of $O_2$, and undergoes multiple $RO_2 + NO \rightarrow R'O_2$ reactions before eventually forming $HO_2$. Such autoxidation chemistry has the net effect of reducing the rate of $RO_2 \rightarrow HO_2$ propagation, and effectively extends the lifetime of $RO_2$ radicals, resulting in higher concentrations. Based on the results in the present work, it is possible that similar chemistry occurred during the ICOZA campaign, although it is unlikely that a BVOC was involved because of the low biogenic influence at the WAO site. However, aromatic species, more relevant to ICOZA, have also been shown to undergo autoxidation (Wang et al., 2017; Mehra et al., 2020).

## 5 Conclusions

OH, $HO_2$, and $RO_2$ radicals and OH reactivity ($k'_{OH}$) were measured at a UK coastal receptor site during the July 2015 ICOZA intensive field campaign. Maximum measured daily OH, $HO_2$, and total $RO_2$ radical concentrations were in the range 2.6–17 $\times 10^6$, 0.75–4.2 $\times 10^8$, and 2.3–8.0 $\times 10^8$ molecule $cm^{-3}$, respectively. $k'_{OH}$ ranged from 1.7 to 17.6 $s^{-1}$ with a median value of 4.7 $s^{-1}$. ICOZA data were split by wind direction to assess differences in the radical chemistry between air that had passed over the North Sea (NW–SE sectors) or over major urban conurbations such as London (SW sector). A PSS calculation underpredicted daytime OH in NW–SE air by ~35% on average, whereas agreement was found within instrumental uncertainty (~26% at $2\sigma$) in SW air. The OH levels predicted by a box model using MCM chemistry were in better agreement with the measurements. However, for $HO_2$, the base MCM model overpredicted the observations in NW–SE air in the afternoon by a factor of ~2–3, whereas reasonable agreement was found for $HO_2$ in SW air when the model was constrained to measured





carbonyls (HCHO, MVK+MACR). In contrast, for total $RO_2$, the model severely underpredicted the observations in both NW–SE and SW air, with measurement/model ratios ranging from ~2–5 in the afternoon to almost 40 in the early morning. The model predicted that the dominant $RO_2$ species in both wind sector types was $CH_3O_2$. $k'_{OH}$ observations were underpredicted by ~34% and ~37% in NW–SE and SW air, respectively.

Measured OH levels were well correlated with $J(O^1D)$, where power law fits to measured OH vs $J(O^1D)$ yielded power
terms similar to previous coastal field campaigns. Good correlations were also observed between measured total $RO_2$ and measured $HO_2$, and the fit slopes indicated that the $RO_2$:$HO_2$ ratio was close to 1:1 in NW–SE air and ~2:1 in the more polluted SW air. The slopes of modelled $RO_2$ vs modelled $HO_2$ were different between day and nighttime data, which was not seen in the observations.

Measured radical and $k'_{OH}$ levels and measurement-to-model ratios displayed strong dependences on NO mixing ratios.
For OH, the PSS calculation could capture the observations at high NO (> 1 ppbv), but underpredicted the observations at low NO (< 200–300 pptv) by a factor of ~2–3, suggesting missing OH sources. The MCM-base model performed better in terms of reproducing the observed dependence of OH on NO but there was still a tendency towards underprediction at low NO. The MCM-base model overpredicted $HO_2$ concentrations at low NO in NW–SE air by a factor of ~3, whereas in SW air, the measurements and model results were in agreement across the full NO range. For $RO_2$, measurement-to-model ratios scaled
with NO, from ~2–3 for NO < 100 pptv to ~10–30 for NO > 1 ppbv, a trend found in all previous field campaigns in which $RO_2$ was measured using the $RO_x$LIF technique. This suggests that peroxy radical chemistry is not well understood under high $NO_x$ conditions. Missing OH reactivity, i.e., the difference between measured and modelled $k'_{OH}$, also scaled with NO. The strong correlation of missing OH reactivity with several OVOCs suggests that the missing reactivity was due to unmeasured VOC oxidation products that were not well simulated by the model, rather than a primary VOC species (e.g., a BVOC).

In our companion paper (Woodward-Massey et al., 2022), we analyse the experimental radical budgets, which are derived from measured quantities only (Tan et al., 2018a), and discuss the main sources and sinks of total $RO_x$ and OH radicals. In addition, we calculate *in situ* ozone production rates from our measurements of peroxy radicals and compare them to those predicted by the MCM model.

**Data availability**

The data used in this study are available from the corresponding authors upon request (l.k.whalley@leeds.ac.uk and d.e.heard@leeds.ac.uk) and are also archived on CEDA (https://archive.ceda.ac.uk/).

**Author contributions**

WJB was the principal investigator of the ICOZA project and was responsible for organisation of the Weybourne field intensive. RWM, LKW, DRC, TI, and DEH were responsible for measurements of radicals, OH reactivity, HCHO, and



photolysis frequencies (*J* values also provided by RS and PSM). LRC, LJK, and WJB made measurements of HONO and aerosol surface area. JL and CR measured NO, $NO_2$, and HONO. BJB was responsible for the long-term operation of the Weybourne Atmospheric Observatory and provided $O_3$, CO, and HCHO data. GLF was responsible for measurements of VOCs. RS and SC developed the AtChem modelling framework. RS conducted the MCM model simulations. RWM, LKW, and RS analysed the data. RWM wrote the manuscript with input from all co-authors.

### Acknowledgements


We thank the science team of the ICOZA project. RWM and DRC are grateful to the NERC for funding PhD studentships. RWM, LKW, DRC, TI, and DEH would like to thank the University of Leeds electronic and mechanical workshops. RWM is grateful to Hans Osthoff (University of Calgary) for the provision of Igor functions, and to Chunxiang Ye (Peking University) and Samuel Seldon (University Of Leeds) for useful discussions. We thank Stephen Ball and Roland Leigh (University of

Leicester) for assistance with the spectral radiometer, and Lloyd Hollis (University of Leicester) for assistance with the chemical ionisation mass spectrometer.

### Financial support

This research has been supported by the NERC (grant nos. NE/K012029/1, NE/K012169/1, and NE/K004069/1).

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





**Tables**




**Table 1.** List of species observed and their corresponding measurement techniques for the ICOZA campaign. For descriptions of simple and complex $RO_2$, see Section 2.1.2. For some species (e.g., NO, $NO_2$, HONO, HCHO) more than one measurement technique was available.

| Observation(s) | Technique | Sampling platform | Institution | Reference(s) |
|---|---|---|---|---|
| OH, $HO_2$, total $RO_2$, "simple" and "complex" $RO_2$ | Fluorescence assay by gas expansion (FAGE) | FAGE container roof | Leeds | Whalley et al. (2013); Whalley et al. (2018); Woodward-Massey et al. (2020) |
| OH reactivity | Laser flash photolysis–laser-induced fluorescence spectroscopy (LFP-LIF) | FAGE container roof | Leeds | Stone et al. (2016) |
| $J(O^1D)$ | Filter radiometry | FAGE container roof | Leeds | Bohn et al. (2016) |
| Photolysis frequencies | Spectral radiometry (two instruments) | FAGE and Leicester containers | Leeds/Leicester | Bohn et al. (2008) |
| HCHO | Laser-induced fluorescence (LIF) | WAO manifold | Leeds | Cryer (2016) |
| HONO | Long path absorption photometry (LOPAP) | Birmingham van roof | Birmingham | Heland et al. (2001); Crilley et al. (2021) |
| Aerosol surface area | Aerodynamic particle sizer (APS) | Birmingham van roof | Birmingham | Chen et al. (1985) |
| $Cl_2$/$ClNO_2$ | Chemical ionisation mass spectrometry | Leicester container roof | Leicester | Sommariva et al. (2018) |
| NO ($NO_2$) | Chemiluminescence (LED $NO_2$ converter) | WAO roof | York | Lee et al. (2009a) |
| $NO_2$ | Cavity-attenuated phase-shift spectroscopy (CAPS) | WAO manifold | York | Kebabian et al. (2008) |
| HONO | Differential photolysis with chemiluminescence detection of NO | WAO roof | York | Reed et al. (2016) |
| $O_3$ | UV absorption | WAO manifold | UEA | - |
| CO | MgO reduction with UV detection | WAO manifold | UEA | Robbins et al. (1968) |
| HCHO | Hantzsch colourimetry | WAO manifold | UEA | Nash (1953) |
| VOCs (up to $C_6$ alkanes/alkenes, acetylene, benzene, toluene) | Gas chromatography with flame ionisation detection (GC-FID) | WAO roof | UEA | - |
| VOCs ($C_8$/$C_9$ aromatics, Σmonoterpenes), OVOCs (methanol, acetaldehyde, acetone, acetic acid, MVK+MACR[a], MEK[b]), acetonitrile, DMS[c] | Proton transfer reaction–mass spectrometry (PTR-MS) | WAO roof | UEA | Murphy et al. (2010) |

[a]Sum of methyl vinyl ketone and methacrolein.

[b]Methyl ethyl ketone.

5  [c]Dimethyl sulfide.





**Table 2.** Median trace gas mixing ratios and meteorological parameters, split according to wind direction (NW–SE = <165° and >285°; SW = 180°–270°).

| Species or meteorological parameter | NW-SE air | SW air |
|---|---|---|
| NO (pptv) | 142 | 233 |
| NO$_2$ (pptv) | 1854 | 2766 |
| HONO (pptv) | 52 | 97 |
| CO (ppbv) | 113 | 106 |
| O$_3$ (ppbv) | 39 | 31 |
| HCHO (pptv) | 925 | 1127 |
| Isoprene (pptv) | 27 | 34 |
| MVK+MACR (pptv) | 40 | 88 |
| PM$_{2.5}$ (µg/m$^3$) | 3.8 | 4.4 |
| Temp (°C) | 15.7 | 17.4 |
| RH (%) | 82 | 72 |
| $J$(O$^1$D) (s$^{-1}$)[a] | $1.7 \times 10^{-5}$ | $1.4 \times 10^{-5}$ |

[a]Reported as the maxima in Figure 4.





**Table 3.** Fit parameters obtained from power law fits ($y = y0 + Ax^{pow}$) to the OH vs $J(O^1D)$ data shown in Figure 9, split according to wind direction (NW–SE = <165° and >285°; SW = 180°–270°).

|  | *NW-SE* | | *SW* | |
| --- | --- | --- | --- | --- |
|  | *Observed* | *MCM-base* | *Observed* | *MCM-base* |
| y0 | $1.2 \times 10^5$ | $1.4 \times 10^4$ | $1.8 \times 10^5$ | $1.0 \times 10^5$ |
| A | $7.2 \times 10^{11}$ | $3.7 \times 10^9$ | $1.1 \times 10^{10}$ | $5.2 \times 10^{10}$ |
| pow | 0.86 | 0.61 | 0.67 | 0.80 |

**Table 4.** Fit parameters obtained from linear fits (y = mx + c, with number of points, N, and Pearson correlation coefficient, R) to the $RO_2$ vs $HO_2$ data shown in Figure 10, split according to wind direction (NW–SE = <165° and >285°; SW = 180°–270°).

|  | *NW-SE* | | *SW* | |
| --- | --- | --- | --- | --- |
|  | *Observed* | *MCM-base* [a] | *Observed* | *MCM-base* [a] |
| m | 1.3 | 0.67 | 1.8 | 0.63 |
| c | $1.6 \times 10^8$ | $9.6 \times 10^6$ | $1.20 \times 10^8$ | $-3.50 \times 10^6$ |
| N | 373 | 113 | 451 | 122 |
| R | 0.63 | 0.96 | 0.81 | 0.97 |

[a]Nighttime data not included.





**Figures**

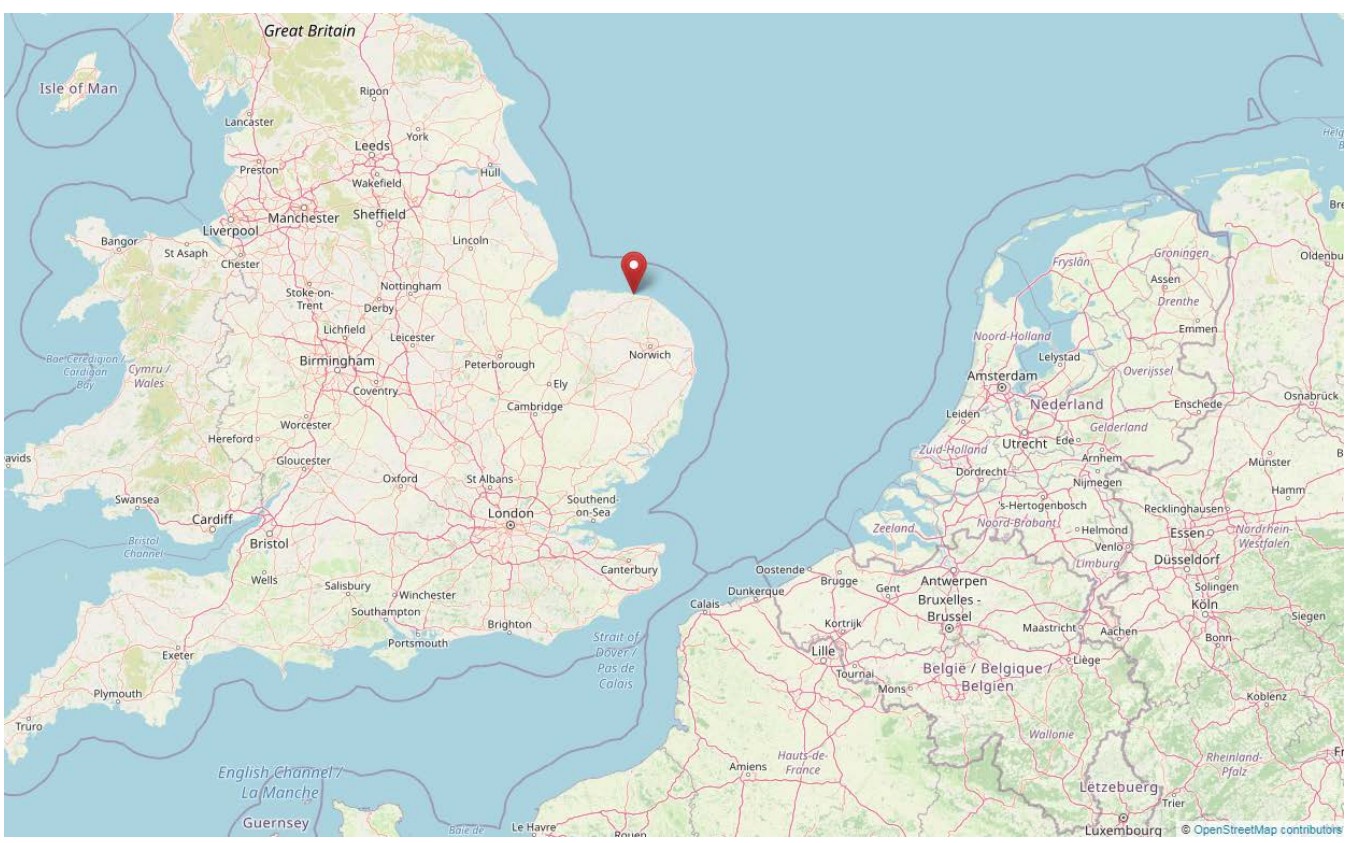

**Figure 1.** Location (red pin) of the Weybourne Atmospheric Observatory (WAO) on the North Norfolk coast (52°57'02" N, 1°07'19"E, 16 m above sea level). © OpenStreetMap contributors 2022. Distributed under the Open Data Commons Open Database License (ODbL) v1.0.





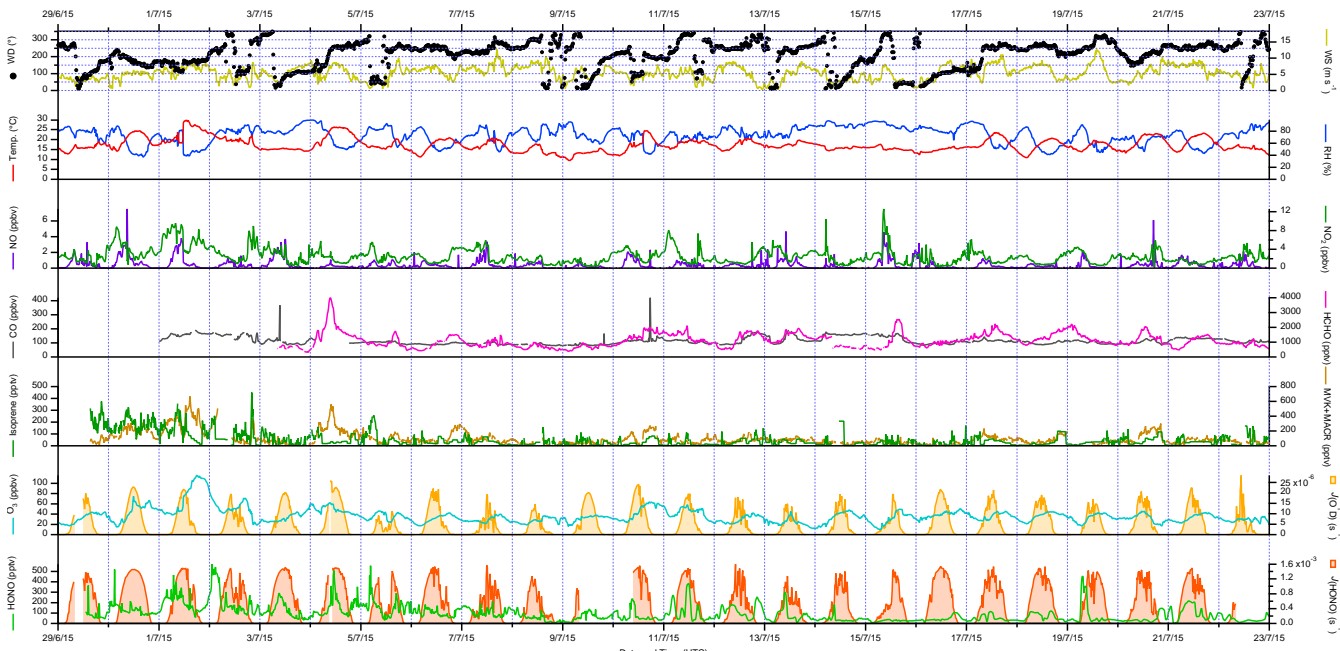

**Figure 2.** Time series of meteorological parameters (wind speed and direction, temperature, RH, photolysis frequencies) and trace gases (NO, NO₂, CO, HCHO, isoprene, MVK+MACR, O₃, HONO) measured during ICOZA (29th June – 23rd July 2015). All data presented are 15 min averages. UTC = Universal Time Coordinated.



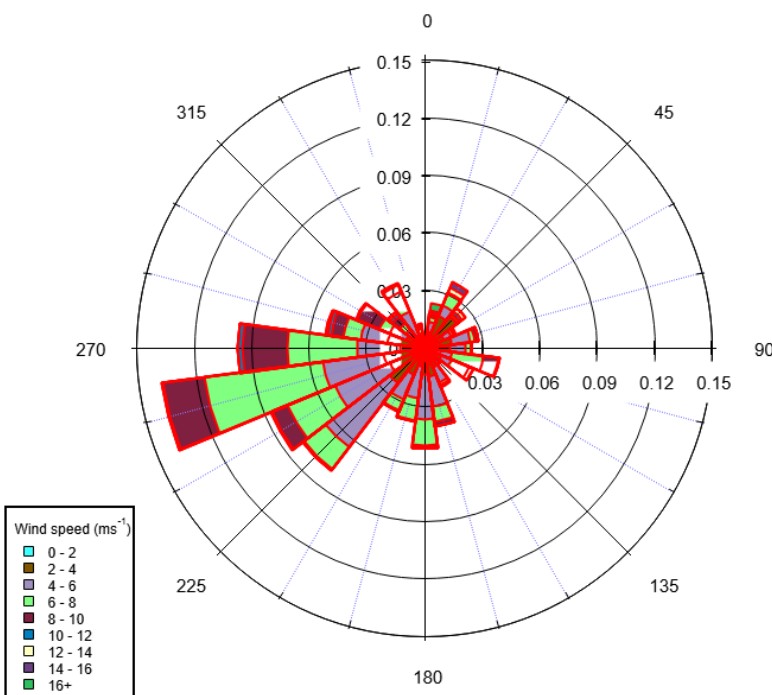

**Figure 3.** Wind rose plot for the entire campaign. Air originated from the SW sector (180°–270°) for approximately 50% of the time.





**Figure 4.** Diel profiles of trace gases and meteorological parameters, split according to wind direction (NW–SE = <165° and >285°; SW = 180°–270°). Data were only included if radical measurements were also available. Shaded areas correspond to 25th and 75th percentiles of the data in each time bin.





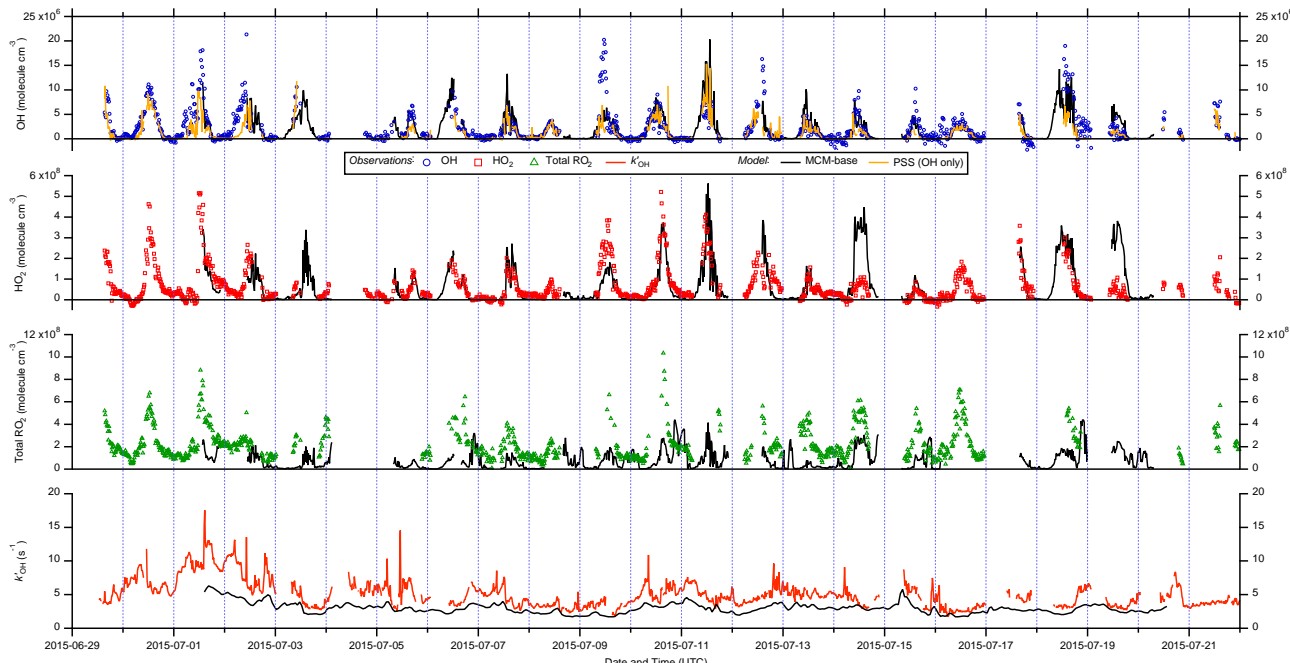

**Figure 5.** Time series of OH, HO$_2$, and total RO$_2$ measurements and comparison to MCM-base model and photostationary steady state (PSS) predictions. All data are at 15 min time resolution except for model OH reactivity (1 h). Error bars omitted for clarity.





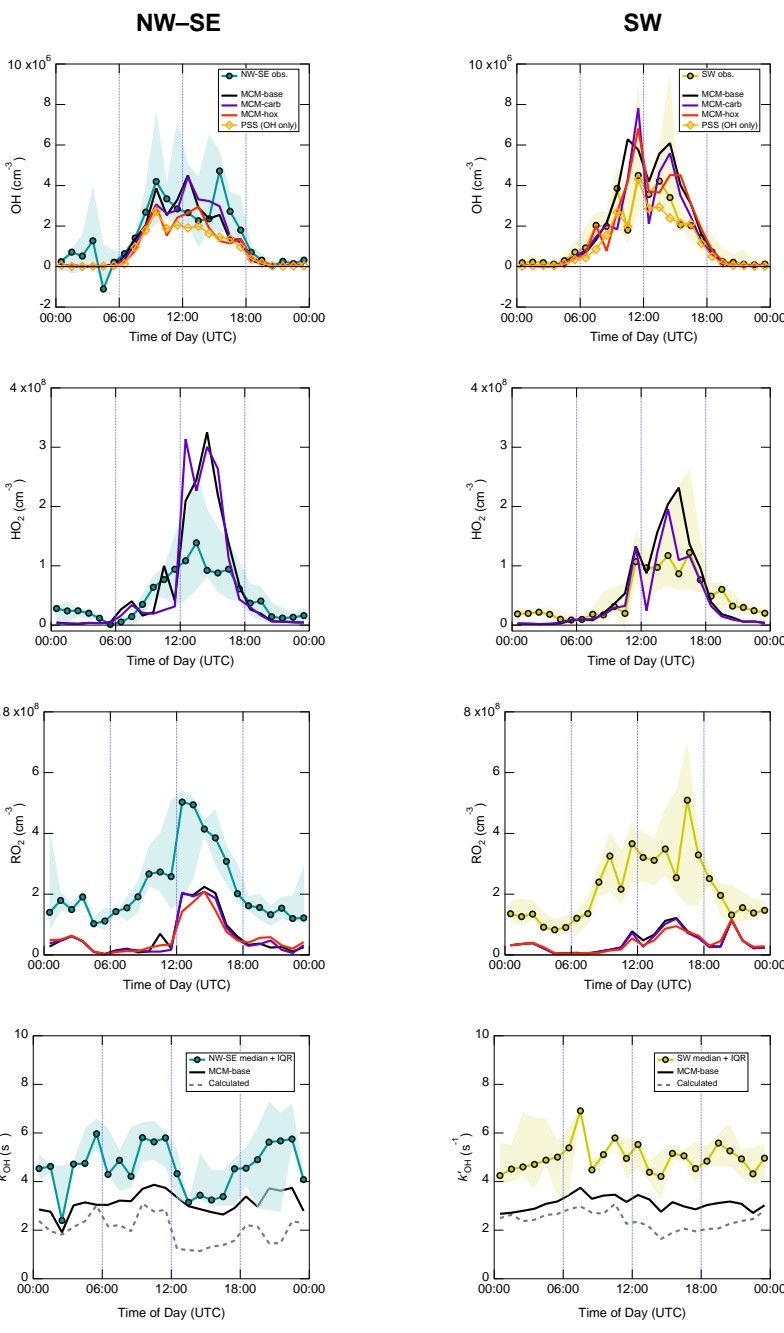

**Figure 6.** Hourly median diel profiles of OH, HO₂, total RO₂, and $k'_{OH}$ and comparison to MCM-base and PSS model predictions, split according to wind direction (left NW–SE, right SW). Shaded areas correspond to 25th and 75th percentiles of the data in each time bin.





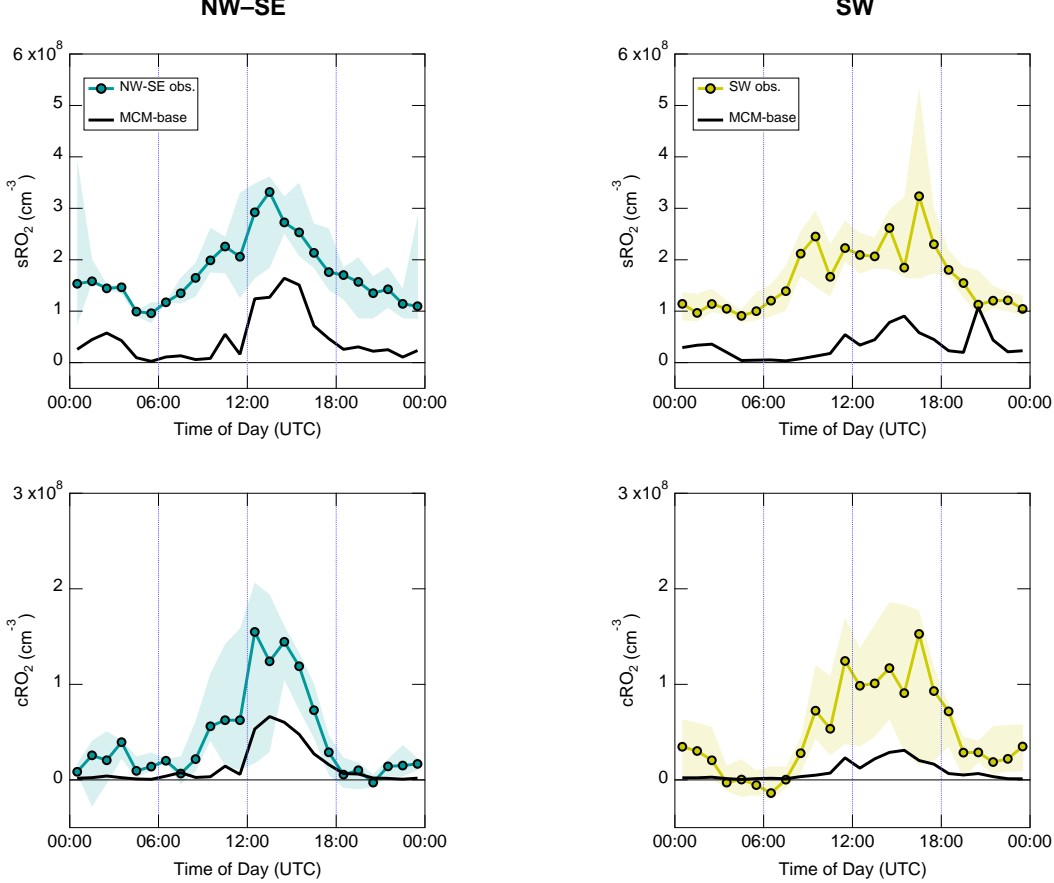

**Figure 7.** Hourly median diel profiles of "simple" ($sRO_2$) and "complex" $RO_2$ ($cRO_2$), defined in text, and comparison to MCM-base model predictions, split according to wind direction (left NW–SE, right SW).





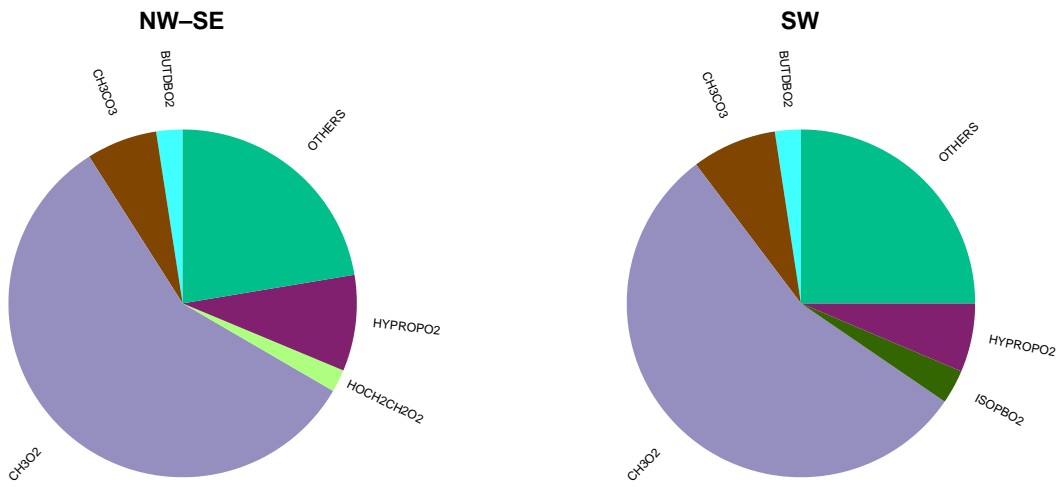

**Figure 8.** Daytime median RO$_2$ contributions predicted by the MCM-base model, split according to wind direction (left: NW–SE, right: SW). For clarity, only the top five RO$_2$ species are shown, otherwise the RO$_2$'s are lumped together as "others". MCM nomenclature used, see text for definitions (Section 3.2).

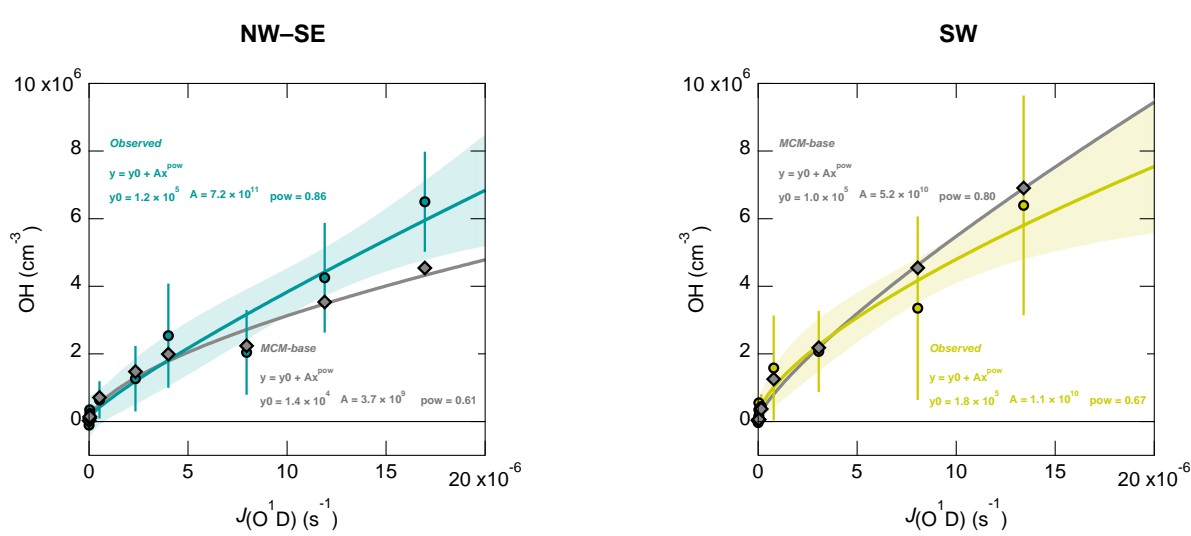

**Figure 9.** Observed and modelled OH binned against $J(O^1D)$ using 12 bins with approximately the same number of points, split according to wind direction (left NW–SE, right SW). Data are shown as means, where error bars on measurement points correspond to one standard deviation (SD, not shown for model data for clarity). Solid lines are power law fits to OH vs $J(O^1D)$, and shaded areas give 95% confidence intervals of the fits to observed data (not shown for model data for clarity).





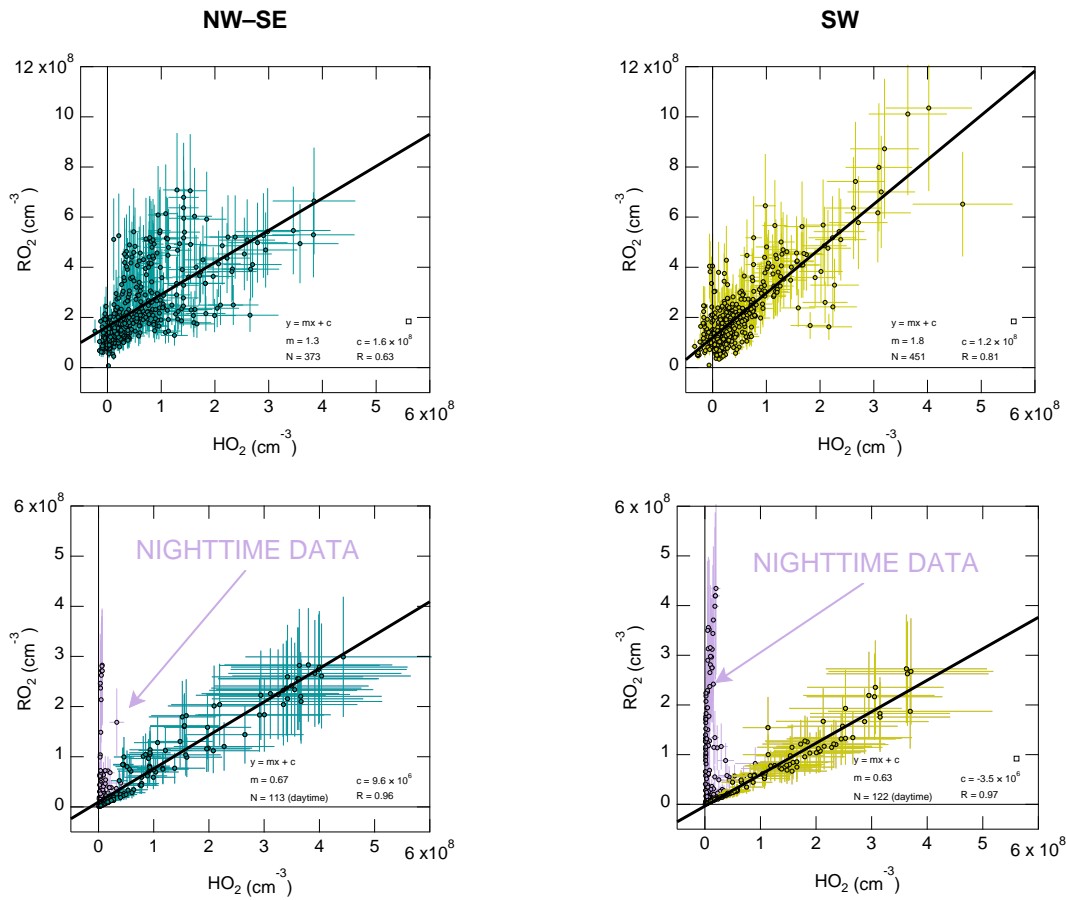

**Figure 10.** Observed total RO$_2$ vs observed HO$_2$ (top) and modelled total RO$_2$ vs modelled HO$_2$ (bottom), split according to wind direction (left NW–SE, right SW). Solid black lines correspond to linear least squares fits. For the model results, nighttime data exhibit a different RO$_2$ vs HO$_2$ slope, highlighted in purple; these data were not included in fits.





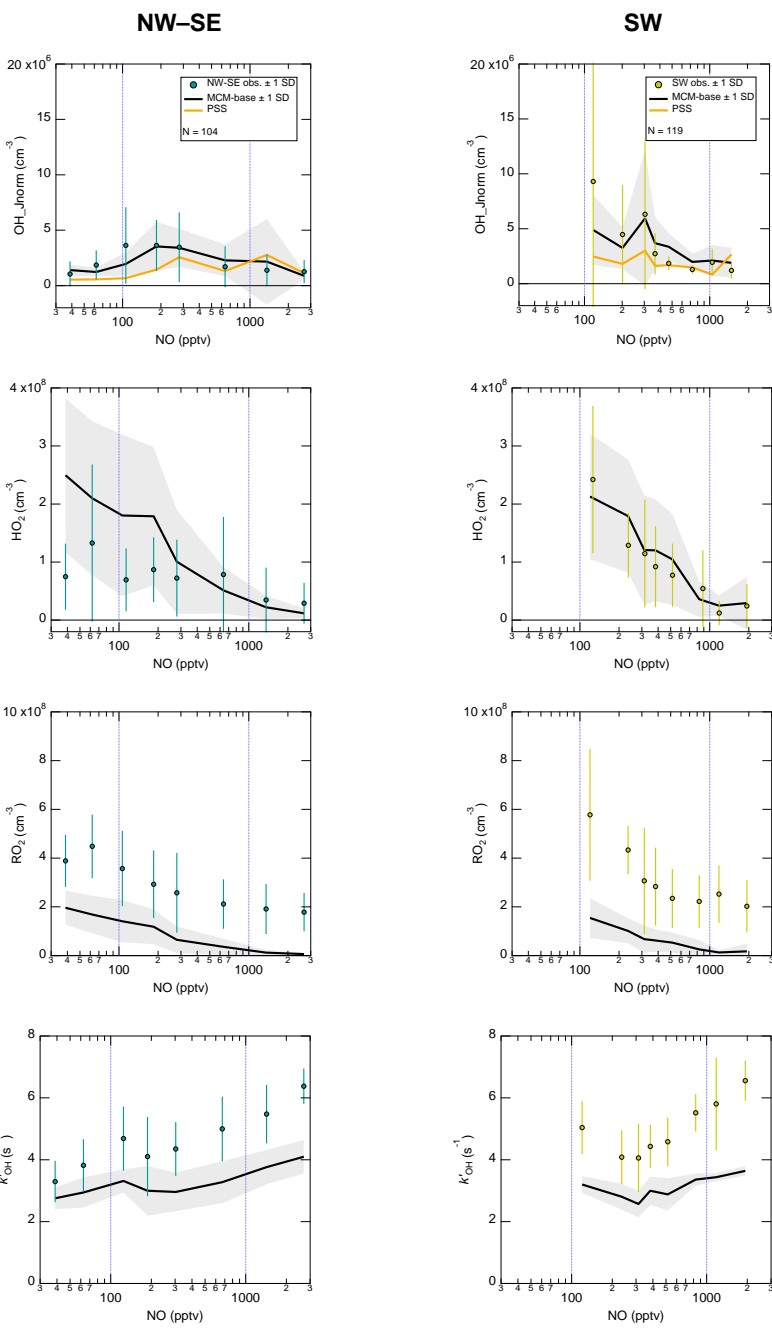

**Figure 11.** Dependence of observed and modelled OH (normalised to average $J(\text{O}^1\text{D})$), HO$_2$, total RO$_2$, and OH reactivity on NO mixing ratios. Only daytime data were included, using the filter $J(\text{O}^1\text{D}) > 5 \times 10^{-7}$ s$^{-1}$. Data are shown as means ± 1 SD, and split according to wind direction (left NW–SE, right SW). Data were separated into 8 NO bins with approximately the same number of points.





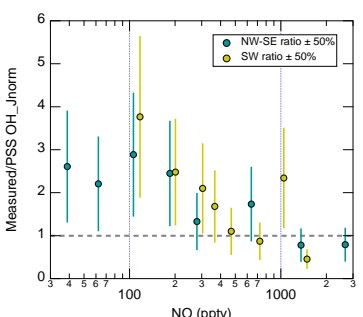 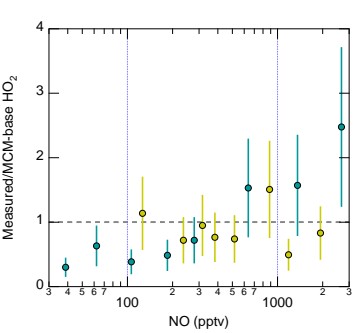 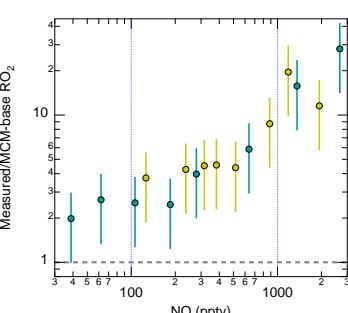

**Figure 12.** NO-dependence of the measurement-model ratios for radical species from Figure 11. Error bars correspond to an estimated combined measurement-model error of 50%. For OH, the reference model is the PSS calculation, and for $HO_2$ and $RO_2$ this is MCM-base. Note the $y$-log scale for $RO_2$.

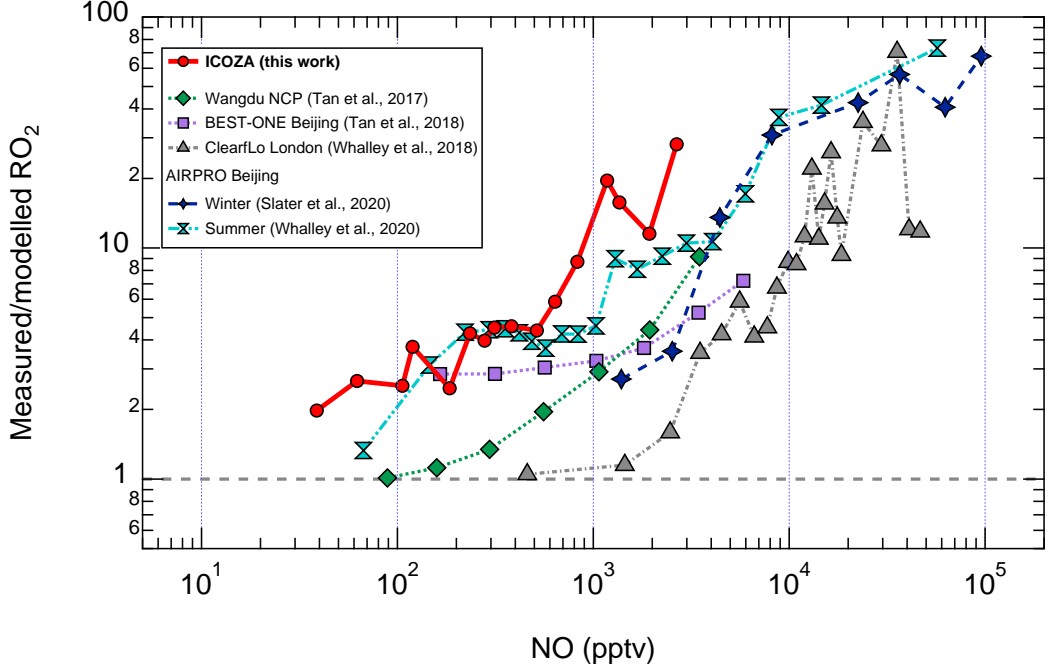

**Figure 13.** Comparison of $RO_xLIF$-measured $RO_2$ measurement-model ratios as a function of NO.





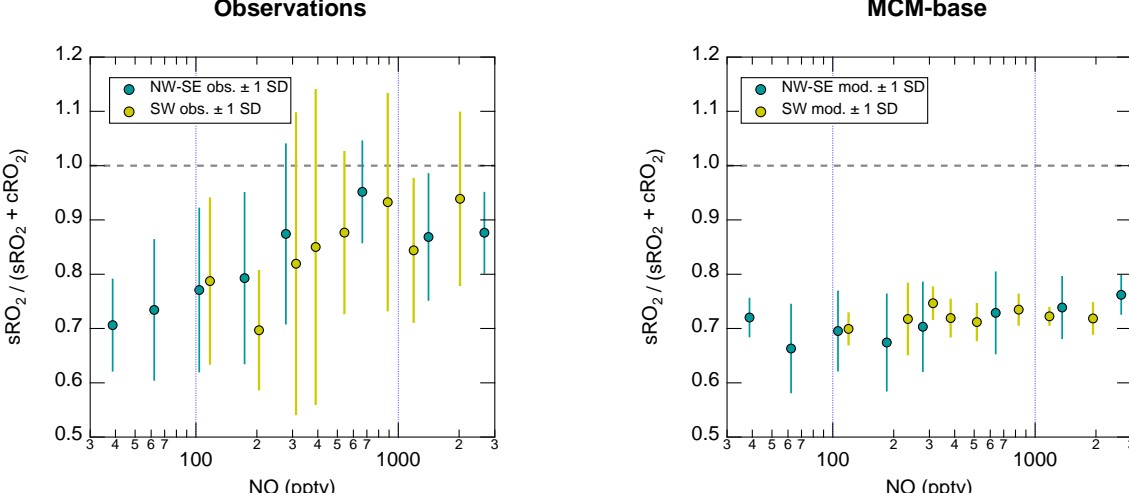

**Figure 14.** Contribution of $sRO_2$ to total $RO_2$ (= $sRO_2$ + $cRO_2$) as a function of NO (measurements left, model results right). Data are shown as means ± 1 SD. Data were separated into 8 NO bins with approximately the same number of points.



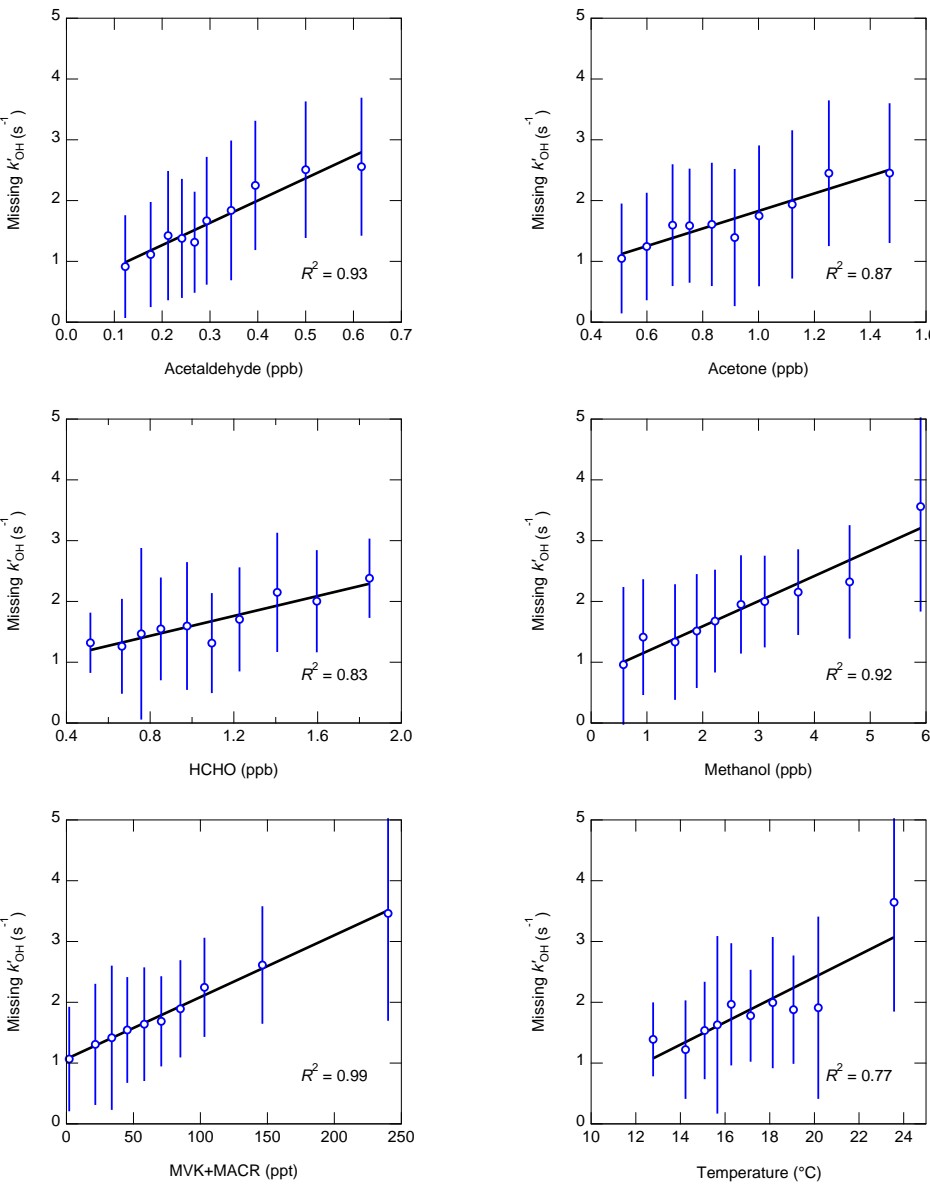

**Figure 15.** Dependence of missing OH reactivity on acetaldehyde, acetone, HCHO, methanol, MVK+MACR, and temperature. Data are shown as means ± 1 SD. In each plot, data were separated into 10 $x$-bins with approximately the same number of points.