# Peer review of "Radical chemistry at a UK coastal receptor site – Part 1: observations of OH, HO2, RO2, and OH reactivity and comparison to MCM model predictions"

_Atmospheric Chemistry and Physics, 2022_

## Author Comment (AC1)

Atmos. Chem. Phys. Discuss., referee comment RC1
https://doi.org/10.5194/acp-2022-207-RC1, 2022
**Comment on acp-2022-207**
Anonymous Referee #1

Referee comment on "Radical chemistry at a UK coastal receptor site - Part 1: observations of OH, HO$_2$, RO$_2$, and OH reactivity and comparison to MCM model predictions" by Robert Woodward-Massey et al., Atmos. Chem. Phys. Discuss., https://doi.org/10.5194/acp-2022-207-RC1, 2022

*We thank the referee for their careful reading of the manuscript and for their helpful suggestions. Our responses are given in italics below each comment, together with any changes to the text. The main change is that the two papers have been merged into one paper, and a lot of material has been moved into the SI.*

The authors report measurements of radicals at a coastal site in the UK. In general, campaigns with a full set of radical measurements are sparse, so that further exploration of radical concentrations in different chemical conditions are valuable. The paper has a companion paper investigating the chemical budgets of radicals. In this manuscript the authors focus on the description of measurements and model-measurement comparisons. A large part of the manuscript is very descriptive, also in the discussion part, which puts the results into the context of results from other campaigns reported literature. Little new results are shown in the sense of improving the understanding radical chemistry in the atmosphere. Therefore, this manuscript rather fits a measurement report instead of a research article. It should be considered to change the manuscript category.

*We have now merged the Part 1 and Part 2 companion papers into a single merged paper, with quite a bit of material moved to the SI. We feel that combining the results of the two papers does provide new understanding of the chemistry of the atmosphere over a range of NOx, and combines a unusually broad range of measurements of radicals (OH, HO2, RO2), OH reactivity with full supporting measurements to constrain the model and to quantify the radical budgets.*

The authors need to improve the manuscript by a concise writing. It is not clear, if separte papers for the measurements / model reszlts and chemical budget would have been required, if the authors had carefully planned to focus the writing of new findings and had cut on lengthy descriptions of figures that can be easily grasped by seeing the figures. In addition. there are some sections, in which it is not clear, if there is a deeper meaning of the analysis that is shown or if the analysis has only be done, because a similar analysis has been done in other papers and therefore, these sections could have been omitted. The manuscript as it is written

now clearly suffers from having the interpretation of model results and of the chemical budgets separated in 2 papers due to the close connection between both. Specifically the PSS calculations for OH shown in this paper is essentially the same as doing a chemical budget presented in the other paper. Merging the 2 papers would clearly be the best to present the insights into radical chemistry and likely also possible, because parts of the papers are similar, since the same data set is analysed and results from each paper is described in the other paper, and descriptive and unnecessary parts can be omitted.

*On reflection we feel that separate papers are not warranted for the radical measurements/model comparisons and the radical budget, for the reasons outlined by the referee. Hence we have merged the two papers, and with conciseness in mind, have reduced the material considerably looking for overlaps between the two papers, and have moved a lot of material into the SI to support the results in the main paper.*

The presentation quality of figures also needs considerable improvements. Font sizes in most of the figures contain are too small to be readable and scaling of axis are not appropriate. Light colours of text as used in the current figures are not suitable for reading (e.g. yellow).

*Where possible we have resized some of the figures to mitigate against this, and where possible altered the figures, and moved quite a few figures to the SI where the figures are presented at a considerably larger size.*

Additional specific comments:

L50: For this type of paper, just showing H-abstraction to form RO2 may oversimplifying the chemistry. Overall, some of the text-book like introduction may not be required.

*We have removed a considerable amount of the text-book like material, in that way it should then not be considered selective via omission of e.g. addition processes.*

L84: I assume that you mean "their photolysis can also be important radical sources"

*Yes that is what is meant. We have used the suggested rewording.*

L94: The conclusion in Novelli et al. is not that Criegee intermediates are the reason for interferences observed in the field, because reactant concentrations in their work were much higher than atmospheric concentration.

*We have removed this reason for the observed interferences.*

L144: Are you sure that the purity of NO was 99.95%? Typically the best purity that is available is only 99.5%.

*Checking the BOC specifications, the purity is now quoted as N2.8, which is 99.8%. This has been changed in the materials section.*

Fig. 2 It may be a good idea to improve visibility by splitting the figure into 2 panels by time.

*We feel that the data being shown all together provides an overview of the campaign in one panel. We have now positioned the figure so it is landscape on a single page so easier to see.*

L330: The statement about HONO is rather short and does not really reflect the high variability that is observed. On some days, values during the day were even higher than during the night.

*We agree that HONO is quite variable – and have added a statement to that effect.*

L384: Looking at the entire time series, the second peak that appears in the median diel profile looks more like an artefact of the median calculation than a real feature of the diel profile as it sounds in this statement.

*We agree, it is just a single point on the median diel profile. We have removed the sentence regarding the second peak.*

L411: I would avoid giving information in the text that is repeating what can be seen in the legend of the figure

*We have removed the repetition, and tried to do this where this may occur elsewhere.*

Fig 7: Here, it may make sense to have the same scale of the y-axis for sRO2 and cRO2.

*There is quite a difference between sRO2 and cRO2 (factor of 3) and we feel that some detail will be lost if the y-axes are the same.*

Fig. 8: Labels of the pie chart are not easy to read. Names may need further explanation in the figure caption. Numbers of fraction could be useful as well as the total median RO2 concentration.

*We have made these bigger and defined them in the caption (using the definitions in Section 3.2 of Paper 1). We have also given the total median RO2 in the caption for the 2 wind directions. We feel the fractions can be estimated by eye using the figure, and the numerical values are given in the text.*

L430 ff: Is the relative abundance of specific RO2 radicals consistent with measured OH reactants? This should be discussed.

*That is a good idea but it is very difficult for some RO2 because they can have multiple sources. Even for the simplest CH3O2, this may derive from several VOCs, not just CH4 but*

*following further degradation of larger RO2. We do indicate the major VOCs which the RO2 derive from, but in this section.*

Section 3.2: RO2: It looks like an offset between measured and modelled simple RO2. Can you exclude that there is an unaccounted instrumental offset?

*We can never completely exclude the possibility of an unaccounted instrumental offset, but our laboratory measurements for a range of VOCs (as described in Whalley et al., 2013) did not reveal any offset. We have assumed that all RO2 are converted to HO2 in the ROxLIF reactor and then to OH in FAGE fluorescence cell (following NO addition) with the same efficiency as CH3O2. If other RO2 species convert less efficiently, this would mean we are underpredicting the total RO. The only way that the measurement can overestimate RO2 is if something is decomposing to form RO2 in the instrument. – Fuchs et al. (2008) estimated HO2NO2 to contribute 1.7% of the measured HO2, CH3O2NO2 contributes 6% to RO2 (assuming an ambient NO2 concentration of 10 ppbv). For PAN, Fuchs et al. (2008) expect 0.1 pptv RO2 per ppbv PAN.*

*H. Fuchs, F. Holland and A. Hofzumahaus, Measurement of tropospheric RO2 and HO2 radicals by a laser-induced fluorescence instrument, Review of Scientific Instruments, 79, 084104 (2008).*

*Whalley, L. K., Blitz, M. A., Desservettaz, M., Seakins, P. W., and Heard, D. E.: Reporting the sensitivity of laser-induced fluorescence instruments used for $HO_2$ detection to an interference from $RO_2$ radicals and introducing a novel approach that enables $HO_2$ and certain $RO_2$ types to be selectively measured, Atmos. Meas. Tech., 6, 3425–3440, https://doi.org/10.5194/amt-6-3425-2013, 2013.*

Section 3.3: This analysis does not give much insights as it is done here. More discussion and comparison with previous findings with interpretation of different and similar results would be needed.

*There is a comparison with previous findings later in Section 4.1. We have signposted this from section 3.3. Also, combining the two papers will help – as the budget analysis for the radicals highlights processes that may be missing – which links to the model-measurement comparison.*

Section 3.4: Again, there is little interpretation or discussion of the correlation and it is not clear what is learned from this analysis. What is the meaning of the different slopes? What is expected for what reason?

*It is quite unusual to see plots of RO2 versus HO2 owing to the scarcity of simultaneously measured data so these correlation plots, and the corresponding ones for modelled values are quite novel. However, we agree there could be more discussion, for example regarding the modelled correlation plot at night, which is very different. Also, combining the two papers into one allows the budget analysis for HO2 and RO2, as RO2 degradation provides a source of HO2.*

L465: The offset does not necessarily indicate that some RO2 sources (= type of RO2 radicals) do not form HO2 as it sounds in the statement. It can also be that the lifetimes of HO2 and RO2 were much different or RO2 loss channels did not lead to HO2 formation, but the RO2 from all sources may still generally form HO2. If there were mainly RO2 sources in the night but little HO2 present, why would you expect that there is a correlation between RO2 and HO2, when the reaction of RO2+NO as most important pathway to HO2 is not relevant in the night?

*We agree and have incorporated the above into the discussion. There is a correlation but the gradient is much greater (RO2:HO2) at night suggesting nighttime RO2 converts to HO2 much less efficiently at night when NO is low, compared with the daytime. However, small amounts of RO2 will be converted to HO2, so a correlation still exists (either because there is a small amount of NO present or the RO2+RO2 self-reaction can form HO2).*

Section 3.6 / Figure 15: Would you expect that an exponential behaviour of BVOC emissions is visible for the range of temperature that is experienced in the campaign? Can you make an estimate, how much RO2 concentrations will change, if you assume additional VOCs in the model to account for the gap between measured and modelled OH reactivity?

*Over the range of T in the plot (13-24 degrees) we would not expect to see an exponential behaviour , but there clearly is a dependence. In the companion budget analysis paper, we discuss the additional RO2 that is made from missing OH reactivity, which is given by missing kOH/DRO2, where DRO2 is the destruction rate of RO2. As we have now merged the two papers, discussion of how RO2 concentrations will change is covered.*

Section 4.4: It would be good to have some numbers of reactive halogen species that would be required to explain observations.

*We state that for Mace Head during NAMBLEX where it was found that up to 40% of HO2 could be lost to IO under low-NOx conditions, for measured IO levels of 0.8–4.0 pptv 675 (Commane et al., 2011). There have been no measurements of IO, HOI or I2, nor BrO, at Weybourne, and there are no exposed sea-weed beds like at Mace Head, and so it is not expected that halogen levels are so high. However, the levels that were seen at Mace Head are the sort of concentrations that would be needed to explain the observations.*

---

## Author Comment (AC2)

Atmos. Chem. Phys. Discuss., referee comment RC2
https://doi.org/10.5194/acp-2022-207-RC2, 2022 ©
Author(s) 2022. This work is distributed under the
Creative Commons Attribution 4.0 License.

**Comment on acp-2022-207**

Anonymous Referee #2

Referee comment on "Radical chemistry at a UK coastal receptor site - Part 1: observations of OH, $HO_2$, $RO_2$, and OH reactivity and comparison to MCM model predictions" by Robert Woodward-Massey et al., Atmos. Chem. Phys. Discuss., https://doi.org/10.5194/acp-2022-207-RC2, 2022

*We thank the referee for their careful reading of the manuscript and for their helpful suggestions. Our responses are given in italics below each comment, together with any changes to the text. The main change is that the two papers have been merged into one paper, and a lot of material has been moved into the SI.*

This paper presents measurements of OH, HO2, RO2, and total OH reactivity at a coastal site during the 2015 ICOZA (Integrated Chemistry of OZone in the Atmosphere) campaign. The authors compare the measurements to predictions by both a photostationary state model as well as a zero-dimensional model based on the Master Chemical Mechanism (MCM 3.3.1). The authors find that in general the MCM model was able to reproduce the measured OH concentrations during the campaign, but overpredicted the measured concentrations of HO2 under lower NOx conditions when air arrived to the site from the northwest-southeast sectors, while underpredicting the measurements when more polluted air arrived to the site from the southwest sector. The authors also found that the model underpredicted the measured RO2 concentrations for both lower and higher NOx air that arrived from all sectors. The authors also find that the measured total OH reactivity was consistently greater than that calculated by the model.

The measurements described add to a growing dataset that suggest that our understanding of radical chemistry under a range of conditions may be incomplete, and as a result are of interest to the atmospheric chemistry community. The results are consistent with several previous measurements, and the authors suggest several possible reasons for the model discrepancies based on these previous results, including missing halogen chemistry and autooxidation of RO2 radicals reducing the rate of conversion to HO2 radicals. Unfortunately, the impact of these proposals on their model results are not included in this paper, as they are discussed in the companion paper. While the companion paper focuses on the impact of their proposed mechanisms on the radical budgets, this paper would benefit from some additional discussion of the impact of the proposed mechanisms on the modeled radical concentrations.

*We have now merged the Part 1 and Part 2 companion papers into a single merged paper, with quite a bit of material moved to the SI. In this way the impact of the proposed*

*mechanisms on both the modelled radical concentrations and also the radical budgets can be combined. There is certainly overlap on the impact on the modelled concentrations and the radical budgets.*

Specifically, the authors should consider including their model results when they reduced the rate of the RO2 +NO propagation rate as discussed in sections 4.4 and 4.5 as it appears that a reduction in this rate, perhaps due to the competition of RO2 autooxidation with radical propagation, improves the agreement with the measured HO2 and RO2 concentrations. While including an expanded discussion of the model results would add to an already lengthy manuscript, the authors should also consider condensing and or moving some of the discussion of previous measurements into a supplement.

*By combining the two papers into a single merged paper, we have been able to condense the material considerably, and have moved a significant amount of material to the SI. The reduced rate means that RO2 will have a longer lifetime and therefore the RO2 concentration will increase in the model, and slower HO2 production via RO2+NO means that the HO2 concentration will decrease – with autoxidation not expected to be a factor under the conditions at Weybourne.*

Additional comments:

1) The authors conducted interference measurements during two different periods, finding that unknown interferences contributed less than 20% to the measured OH signal. It appears that these measurements occurred during both NW-SE and SW periods. Did the authors see a significant difference in the interference measurements from the different wind sectors?

*No, we did not see a significant difference from the different wind sectors.*

2) The authors should consider highlighting the NW-SE and SW periods on Figure 5 to help illustrate the impact of the different air masses on the radical measurements.

*The top panel of Figure 2 shows the wind direction, from which the NW-SE and SW periods can be ascertained (either points high on the plot, or low on the plot). Fig. 5 is quite complex already, and we do not want to complicate it further, as there are quite a few changes in wind direction from NW-SE during the campaign. In the caption to Figure 5 we will point to Figure 2 for information about the time-series of wind direction.*

Given that the main focus of the paper is on the measurement/model discrepancy of the radical concentrations, there are several sections and figures in the paper that could be moved to a supplement to improve readability. In particular, sections 3.3 and 3.4 along with figures 9 and 10 could be moved to a supplement.

*We have merged the two companion papers, and moved significant sections into the SI. Included in the material moved to the SI are the sections mentioned above and the accompanying figures.*

3)  The authors could also condense much of the discussion of previous measurements by including a table summarizing the previous measurements/model agreement under the different NO conditions and referencing the table in the discussion.

*We have constructed a new table which summarises the previous measurements, whilst retaining key information in the text.*

4)  As mentioned above, the authors should consider adding the reduced RO2+NO model results to Figures 5-7 to illustrate how this model improves the agreement with the measurements. This illustration of the impact of the reduced rate is not included in the companion paper.

*A model with reduced RO2+NO was not run. Rather in the second paper the impact of reducing the RO2+NO on the budgets of RO2, HO2 and OH was considered. As the two papers have now been merged, we will merge together the material on reducing the RO2+NO rate, discussing the impact on the budget, and likely impact on their concentrations.*